# SEMI-VARIANCE REDUCTION FOR FAIR FEDERATED LEARNING

## ABSTRACT

Ensuring fairness in Federated Learning (FL) systems, i.e. a satisfactory performance for all of the diverse clients in the systems, is an important and challenging problem. There are multiple fair FL algorithms in the literature, which have been relatively successful in providing fairness. However, these algorithms mostly emphasize on the loss functions of worst-off clients to improve their performance, which often results in the suppression of well-performing ones. As a consequence, they usually sacrifice the system overall average performance for achieving fairness. Motivated by this and inspired by two well-known risk modeling methods in Finance, *Mean-Variance* and *Mean-Semi-Variance*, we propose and study two new fair FL algorithms, *Variance Reduction* (VRed) and *Semi-Variance Reduction* (Semi-VRed). VRed encourages equality between clients loss functions by penalizing their variance. In contrast, Semi-VRed penalizes the discrepancy of only the worst-off clients loss functions from the average loss. Through extensive experiments on multiple vision and language datasets, we show that, Semi-VRed achieves SoTA performance in scenarios with highly heterogeneous data distributions and improves both fairness and system overall average performance.

## 1 INTRODUCTION

Federated Learning McMahan et al. (2017) is a framework consisting of some clients and the private data that is distributed among them, and it allows training of a shared or personalized model based on the clients data. Since the invention of FL by proposing the well-known FedAvg algorithm (McMahan et al., 2017), it has attracted an intensive amount of attention and much progress has been made in its different aspects, including algorithmic innovations (Li et al., 2020b; Reddi et al., 2020a; Pathak & Wainwright, 2020; Huo et al., 2020; Wang et al., 2020; Reddi et al., 2020b; Qu et al., 2022), fairness (McMahan et al., 2017; Li et al., 2020c; Mohri et al., 2019; Li et al., 2020a; Yue et al., 2021; Zhang et al., 2022a), convergence analysis (Khaled et al., 2020; Li et al., 2020; Gorbunov et al., 2021), personalization (Zhang et al., 2021; Chen & Chao, 2022; Oh et al., 2022; Zhang et al., 2022b; Bietti et al., 2022).

Due to heterogeneity in clients data and their resources, performance fairness is an important challenge in FL systems. There have been some previous works addressing this problem. For instance, Mohri et al. (2019) proposed Agnostic Federated Learning (AFL), which aims at minimizing the largest loss function among clients through a minimax optimization framework. Similarly, Li et al. (2020a) proposed an algorithm called TERM using tilted losses. Ditto (Li et al., 2021) is another existing algorithm based on model personalization for clients[1]. Also, $q$-Fair Federated Learning ($q$-FFL) (Li et al., 2020c) is an algorithm inspired by $\alpha$-fairness in wireless networks (Lan et al., 2010). Recently, Zhang et al. (2022a) proposed PropFair based on the concept of Proportional Fairness (PF). Interestingly, they also showed that all the aforementioned fair FL algorithms can be unified into a generalized mean framework. GiFair (Yue et al., 2021) is another recent algorithm which achieves fairness using a different mechanism than the previously mentioned algorithms: by penalizing the discrepancy between clients loss functions, i.e. encouraging equality. FCFL (Cui et al., 2021) uses a constrained version of AFL for achieving both algorithmic parity and performance consistency in FL settings.

---

[1]In order to have fair comparison with our baseline algorithms, we do not use model personalization in this work.

Being designed for fair `FL`, the aforementioned algorithms usually result in the suppression of well-performing clients, due to the lower weights that the algorithms place on them or due to the equality that is encouraged between clients losses (`GiFair`). As a consequence, they achieve an overall average performance which is either smaller than or close to that of vanilla FedAvg. This is our motivation for proposing our two new algorithms.

Our inspiration in this paper is a concept in Finance called *risk modeling* used for portfolio selection. There are two vastly used methodologies for risk modeling: *Mean-Variance* (`MV`) (Zhang et al., 2018; Soleimani et al., 2009; Markowitz, 1952) and its expansion: *Mean-Semi-Variance* (`MSV`) (Boasson et al., 2017; Plà-Santamaria & Bravo, 2013; Ballestero, 2005; Stuart & Markowitz, 1959), which are used for quantifying investment return and investment risk. Motivated by the vast usage of these methodologies and their great success in financial planning, we bring the `MV` and `MSV` methods to `FL` by proposing our *Variance Reduction* (`VRed`) and *Semi-Variance Reduction* (`Semi-VRed`) algorithms, respectively. By conducting extensive experiments on popular vision and language datasets, we show that our `VRed` algorithm achieves a performance competitive to existing baseline fair `FL` algorithms. More importantly, `Semi-VRed` achieves state-of-the-art performance in terms of both fairness and system overall average performance.

## 2 BACKGROUND

With formal notations, we consider an `FL` setting with $n$ clients for the task of multi-class classification. Let $x \in \mathcal{X} \subseteq \mathbb{R}^p$ and $y \in \mathcal{Y} = \{1, \dots, C\}$ denote the input data point and its target label, respectively. Each client $i$ has its own private data with data distribution $P_i(x, y)$. Let $h : \mathcal{X} \times \Theta \to \mathbb{R}^C$ be the used predictor function, which is parameterized by $\theta \in \Theta \subseteq \mathbb{R}^d$ shared among all clients. Also, let $\ell : \mathbb{R}^C \times \mathcal{Y} \to \mathbb{R}_+$ be the loss function, which we choose to be the cross entropy loss. Client $i$ minimizes loss function $f_i(\theta) = \mathbb{E}_{(x,y) \sim P_i(x,y)}[\ell(h(x, \theta), y)]$ with minimum value of $f_i^*$.

There are various fair `FL` algorithms in the literature. In table 3 in the appendix, we have provided details of the most recent algorithms with their formulations. The existing fair `FL` algorithms can be grouped into two main categories:

### 2.1 ALGORITHMS BASED ON THE GENERALIZED MEAN

This category of algorithms includes `FedAvg` (McMahan et al., 2017), `q-FFL` (Li et al., 2020c), `AFL` (Mohri et al., 2019), `TERM` (Li et al., 2020a), `PropFair` (Zhang et al., 2022a). It was shown by Zhang et al. (2022a) that this set of existing fair `FL` algorithms can be unified into a generalized mean framework (Kolmogorov, 1930), where more attention is paid to the clients with larger losses.

### 2.2 ALGORITHMS BASED ON ENCOURAGING EQUALITY

The second category of fair `FL` algorithms, which includes `GiFair`, is based on encouraging equality between clients loss functions. `GiFair` adds a regularization term to the objective of `FedAvg` to penalize the discrepancy between clients loss functions (see table 3 in the appendix). In this way, it encourages equality between clients loss functions.

A common feature of all the aforementioned algorithms is their emphasis on the clients with relatively larger losses, which usually results in the suppression of the well-performing clients. This might result in the degradation of the overall average performance (measured by the mean test accuracy across clients). In the next sections, we will see that `Semi-VRed` can achieve fairness by regularizing the *semi-variance* of clients loss functions and improves both fairness *and* the system overall performance simultaneously. In the context of variance regularization, there have been some works in the literature: Maurer & Pontil (2009); Namkoong & Duchi (2017) propose regularizing the empirical risk minimization (ERM) by the empirical variance of losses across training samples to balance bias and variance and improve out-of-sample (test) performance and convergence rate. Similarly, Shivaswamy & Jebara (2010) propose boosting binary classifiers based on a variance penalty applied to exponential loss. Variance regularization has also been used for out-of-distribution (domain) generalization: assuming having access to data from multiple training domains, Krueger et al. (2021) propose penalizing variance of training risks across the domains, as a method of distributionally robust optimization, to provide domain generalization.

## 3 RISK MODELING METHODS IN FINANCE: *Mean-Variance* AND *Mean-Semi-Variance*

*Mean-Variance* (MV) and *Mean-Semi-Variance* (MSV) have been the most popular methods for Modeling risks and gains of an investment portfolio, which is a first step in financial planning.

***Mean-Variance* (MV)** (Zhang et al., 2018; Soleimani et al., 2009; Markowitz, 1952). This method treats the return of each security in an investment portfolio as a random variable and adopts its expected value and variance to quantify the return and risk of the portfolio, respectively. An investor either minimizes the risk for a fixed expected return level or maximizes the return for a given acceptable risk level. For instance, in the former case, MV results in the following problem:

$$\max_{x_1,\ldots,x_n} \quad \mathbb{E}[x_1 S_1 + \ldots + x_n S_n] \tag{1}$$
$$\text{s.t.} \quad \sigma^2[x_1 S_1 + \ldots + x_n S_n] \leq R, \quad \sum_i x_i = 1, \quad x_i \geq 0.$$

Here $\mathbb{E}$ and $\sigma^2$ denote the expected value and variance operators, respectively. Also, $S_i$ and $x_i$ denote the random return from security $i$ and the proportion of total wealth invested in security $i$, respectively. This example has provided a basic view of how MV model works. Other closely related measures of risk in the MV model include the standard deviation ($\sigma$) and coefficient of variation ($\sigma/\mathbb{E}$). However, the *Mean-Variance* modeling of risk is debatable: any uncertain return above the expectation is usually not considered as risk in the common sense, but the MV model does so. This shortcoming is resolved by the *Mean-Semi-Variance* model.

***Mean-Semi-Variance* (MSV)** (Boasson et al., 2017; Plà-Santamaria & Bravo, 2013; Ballestero, 2005; Stuart & Markowitz, 1959). Having recognized the importance of the (often) one-side nature of risks, MSV model proposed a *downside* risk measure known as *semi-variance*, which we denote by $\sigma^2_<$. Unlike variance, it is only concerned with the downside of the return. i.e. only the cases that the return drops *below* a predefined threshold. With this risk modeling method, problem 1 changes to the following:

$$\max_{x_1,\ldots,x_n} \quad \mathbb{E}[x_1 S_1 + \ldots + x_n S_n] \tag{2}$$
$$\text{s.t.} \quad \sigma^2_<[x_1 S_1 + \ldots + x_n S_n] \leq R, \quad \sum_i x_i = 1, \quad x_i \geq 0,$$

where the operator semi-variance measures the *downsides* of the return: $\sigma^2_<[z] = \mathbb{E}[(\mathbb{E}[z] - z)^2_+]$. MSV is a preferable alternative for the MV model as its modeling of the risk is more consistent with our perception from an investment risk. Again, the problem above gives a basic understanding of how the MSV model works. More complex variations of MV and MSV models have been developed for complex and unpredictable financial markets (Rigamonti & Lučivjanská, 2022; Zhang et al., 2018; Ballestero, 2005).

## 4 MV AND MSV MODELS FOR FAIR FL

In this section, we propose two fair FL algorithms based on the MV and MSV models. We use the two models to quantify the inequality between clients performances. Inspired by Zhang et al. (2022a), we take a simple approach and define $u_i(\theta) = M - f_i(\theta)$ as the utility of client $i$, where $M$ can be treated as a utility baseline. The smaller the loss function of a client becomes, the larger its utility becomes: *the utility of a client can be used to roughly represent the test accuracy of the shared model, parameterized by θ, on its local data*. With this definition, we propose to model the inequality between clients by the variance and semi-variance of their utilities, resulting in the VRed and Semi-VRed algorithm, respectively.

---

**Algorithm 1:** `VRed` and `Semi-VRed`

---

**Input:** global epoch $T$, client number $n$, loss function $f_i$, number of samples $n_i$ for client $i$, , number of total samples $N$, initial global model $\theta_0$, local step number $K_i$ for client $i$, learning rate $\eta$

1 Let $p_i = \frac{n_i}{N}$ for $i \in \{0, 1, \ldots, n-1\}$
2 **for** $t = 0, 1 \ldots T - 1$ **do**
3     randomly select $\mathcal{S}_t \subseteq [n]$
4     $\theta_t^{(i)} = \theta_t$ for $i \in \mathcal{S}_t$, $N = \sum_{i \in \mathcal{S}_t} n_i$
5     **for** $i$ *in* $\mathcal{S}_t$ **do** // `in parallel`
6         starting from $\theta_t^{(i)}$, take $K_i$ local SGD steps on $f_i(\theta_t^{(i)})$ with learning rate $\eta$ to find $\theta_{t+1}^{(i)}$
7         compute $\Delta_i^{(t)} = \theta_t^{(i)} - \theta_{t+1}^{(i)}$
8     compute $\overline{f}(\theta_t) = \sum_i p_i f_i(\theta_t)$ and $\overline{\Delta}^{(t)} = \sum_i p_i \Delta_i^{(t)}$
9     **if** `VRed` **then**
10         compute $\Delta_t = \sum_i p_i \Delta_i^{(t)} + 2\beta \sum_i p_i (f_i(\theta_t) - \overline{f}(\theta_t))(\Delta_i^{(t)} - \overline{\Delta}^{(t)})$
11     **else if** `Semi-VRed` **then**
12         compute $\Delta_t = \sum_i p_i \Delta_i^{(t)} + 2\beta \sum_i p_i (f_i(\theta_t) - \overline{f}(\theta_t))_+ (\Delta_i^{(t)} - \overline{\Delta}^{(t)})$
13     update $\theta_{t+1} = \theta_t - \Delta_t$
**Output:** global model $\theta_T$

---

## 4.1 THE VRED ALGORITHM

`VRed` models the inequality between clients utilities by their variance and aims to minimize the following objective function:

$$\min_\theta \ F(\theta) = \mathbb{E}[\{f_i(\theta)\}_{i=1}^n] + \beta\sigma^2[\{u_i(\theta)\}_{i=1}^n] = \sum_i p_i f_i(\theta) + \beta \sum_i p_i \left(u_i(\theta) - \sum_j p_j u_j(\theta)\right)^2$$

$$= \sum_i p_i f_i(\theta) + \beta \sum_i p_i \left(f_i(\theta) - \sum_j p_j f_j(\theta)\right)^2. \tag{3}$$

This objective, in addition to minimizing the vanilla `FedAvg` mean loss, reduces the variance of clients utilities. Let us derive the `VRed` federated learning algorithm. By taking the gradient of equation 3 and multiplying it by the step size $\eta$, we have:

$$\eta\nabla F(\theta) = \sum_i p_i \eta\nabla f_i(\theta) + 2\beta \sum_i p_i \left(f_i(\theta) - \sum_j p_j f_j(\theta)\right)\left(\eta\nabla f_i(\theta) - \sum_j p_j \eta\nabla f_j(\theta)\right). \tag{4}$$

This equation immediately leads to an `FL` algorithm, by replacing the gradient $\eta\nabla f_i(\theta)$ with the pseudo-gradient (i.e., the opposite of the local update), denoted by $\Delta_i^{(t)}$:

$$\eta\nabla F(\theta) = \sum_i p_i \Delta_i^{(t)} + 2\beta \sum_i p_i \left(f_i(\theta) - \overline{f}(\theta)\right)\left(\Delta_i^{(t)} - \overline{\Delta}^{(t)}\right), \tag{5}$$

where $\overline{f}(\theta) = \sum_i p_i f_i(\theta)$ and $\overline{\Delta}^{(t)} = \sum_i p_i \Delta_i^{(t)}$. The corresponding algorithm is included in algorithm 1. There is a parameter $\beta$ which tunes the effect of the regularization term, which needs to get tuned for better performance. Note that this is a new aggregation rule: instead of simply averaging the local models, it has an additional second term, which relates to the variance of clients losses. If all clients are identical, this term would vanish.

### 4.1.1 AN INTERPRETATION OF VRED

With the definition of utilities in the previous section ($u_i(\theta) = M - f_i(\theta)$), the objective function of `VRed` algorithm (equation 3) is aimed to penalize the variance of clients utilities. One potential drawback of this is that it might result in the suppression of well-performing clients (the ones

with small losses) for reducing the variance, which is the same drawback that `GiFair` (Yue et al., 2021) had. Hence, the final model overall performance averaged across clients might get sacrificed for achieving fairness. In fact, `GiFair` minimizes an upper bound of `VRed` objective function: assuming $p_i = \frac{1}{n}$ $(i = 1, \ldots, n)$, i.e., all clients have the same number of data points, we have the following upper bound on `VRed` objective function (see equation 18 in the appendix for derivation):

$$\sum_i f_i + \beta \sum_i \left| f_i(\theta) - \frac{1}{n} \sum_j f_j(\theta) \right|^2 \leq \sum_i f_i + \frac{2\beta}{n^2} \sum_{j \neq i} \left| f_i(\theta) - f_j(\theta) \right|^2, \qquad (6)$$

where the right hand side is the same as `GiFair` objective function (except the power 2 used for measuring the pairwise distances between clients losses, see table 3 in the appendix). Therefore, `GiFair` in fact minimizes an upper bound of `VRed`'s objective function. This might be the reason explaining why our `VRed` usually outperforms `GiFair` in terms of fairness in our experiments.

In typical `FL` settings, the global objective function can be written as a weighted sum of clients loss functions, i.e. $F(\theta) := \sum_{i=1}^n w_i h_i(\theta)$, where $h_i(\theta)$ is used by client $i$ as a surrogate of the global objective and is optimized using the client local data. Also, the weight $w_i$ represents the importance of client $i$ loss function in the global objective function $F(\theta)$. For example, `FedAvg` simply uses $h_i(\theta) = f_i(\theta)$ and $w_i = p_i$ ($p_i = \frac{n_i}{N}$, see algorithm 1) and q-FFL uses $h_i(\theta) = f_i^{q+1}(\theta)$ and $w_i = p_i$. A direct consequence of the above summation form for $F(\theta)$ is:

$$\nabla F(\theta) = \sum_{i=1}^n w_i \nabla h_i(\theta), \qquad (7)$$

Again, the weight $w_i$ represents the importance of the client $i$ model updates. In lemma 1, we show that the gradient of the global objective of `VRed` in equation 3, can be written in the form of equation 7. *For simplicity and easier interpretation, we assume $p_i = \frac{1}{n}$ $(i = 1, \ldots, n)$, i.e., all clients have the same number of data points.*

**Lemma 1.** *For any model parameter $\theta$, the gradient of the global objective $F(\theta)$ defined in equation 3 can be expressed as*

$$\nabla F(\theta) = \sum_{i=1}^n w_i(\theta) \nabla f_i(\theta), \;\; w_i(\theta) = \frac{1}{n} + \frac{2\beta(f_i(\theta) - \overline{f}(\theta))}{n}, \;\; \overline{f}(\theta) = \frac{\sum_i f_i(\theta)}{n}. \qquad (8)$$

The proof is deferred to §A in the appendix. Lemma 1 shows that, unlike `FedAvg` that would assign $w_i = \frac{1}{n}, i = 1, \ldots, n$ to all clients, `VRed` assigns a relatively larger weight ($w_i$) to clients with larger loss functions, and dynamically updates the weights $w_i$ at each communication round. We will provide an interpretation of this finding about `VRed` in the next sections. Importantly, based on equation 8, in order for all clients to get assigned a positive weight, the parameter $\beta$ needs to satisfy the following inequalities: $0 \leq \beta < \beta_{\mathrm{VRed}}^{max}(\theta) \triangleq \frac{1}{2(\overline{f}(\theta) - \min_i \{f_i(\theta)\})}$.

## 4.2 The Semi-VRed algorithm

Inspired by the discussion on the superiority of `MSV` over `MV` in § 3 for risk modeling, we propose an extension of `VRed`. Consider the following objective function instead of equation 3:

$$\min_\theta F(\theta) = \mathbb{E}[\{f_i(\theta)\}_{i=1}^n] + \beta \sigma_<^2[\{u_i(\theta)\}_{i=1}^n] = \sum_i p_i f_i(\theta) + \beta \sum_i p_i \left( f_i(\theta) - \sum_j p_j f_j(\theta) \right)_+^2 \qquad (9)$$

where $\sigma_<^2$ denotes the semi-variance of clients utilities. This objective, in addition to minimizing the mean loss, *reduces the semi-variance of clients losses*, meaning that only those clients that have relatively small utilities $u_i(\theta)$ (or equivalently large losses $f_i(\theta)$) contribute to the semi-variance regularization term in eq. (9).

Similar to what we did for `VRed`, if we take the gradient of equation 9, we have:

$$\eta \nabla F(\theta) = \sum_i p_i \Delta_i^{(t)} + 2\beta \sum_i p_i \left( f_i(\theta) - \overline{f}(\theta) \right)_+ \left( \Delta_i^{(t)} - \overline{\Delta}^{(t)} \right), \tag{10}$$

where $\Delta_i^{(t)}$ is the pseudo-gradient (i.e., the opposite of the local update) of user $i$. The corresponding algorithm is included in algorithm 1. Again, we have a tunable parameter $\beta$ which sets the effect of the regularization term and needs to get tuned for better performance.

### 4.3 CAN WE INTERPRET WHAT SEMI-VRED DOES?

#### 4.3.1 OPTIMIZATION ASPECT

We will show in lemma 2 that, in contrast to `VRed` (and `GiFair`) and thanks to its more efficient formulation, `Semi-VRed` does not suppress the well-performing clients to help the worst-off ones. *Similar to lemma 1, we assume $p_i = \frac{1}{n}$ $(i = 1, \dots, n)$, for simplicity and easier interpretation.*

**Lemma 2.** *In each communication round between the clients and the server, let $>_C$ denote the set of clients whose local loss function is greater than the average loss function $\overline{f}(\theta)$. For any model parameter $\theta$, the gradient of the global objective $F(\theta)$ defined in equation 9 can be expressed as*

$$\nabla F(\theta) = \sum_{i=1}^n w_i(\theta) \nabla f_i(\theta), \tag{11}$$

*where:*

$$\overline{f}(\theta) = \frac{\sum_i f_i(\theta)}{n}, \quad w_i(\theta) = \begin{cases} \dfrac{1}{n} + \dfrac{2\beta(f_i(\theta) - \overline{f}(\theta))}{n} - \dfrac{2\beta \sum_{j \in >_c}(f_j(\theta) - \overline{f}(\theta))}{n^2}, & \textit{if } i \in >_c \\ \dfrac{1}{n} - \dfrac{2\beta \sum_{j \in >_c}(f_j(\theta) - \overline{f}(\theta))}{n^2}, & \textit{if } i \notin >_c \end{cases} \tag{12}$$

Similar to `VRed`, there is an upper-bound for $\beta$ to ensure positive weights for all clients in equation 12: $0 \leq \beta < \beta_{\text{Semi-VRed}}^{max}(\theta) \triangleq \frac{n}{2 \sum_{j \in >_c}(f_j(\theta) - \overline{f}(\theta))}$.

**Remark 1.** *Interesting points can be observed by comparing lemma 1 and lemma 2. First, both of the algorithms pay more attention to worst-off clients by assigning larger weights to their gradients. However, `Semi-VRed` assigns relatively larger weights to the well-performing clients. Also, for `VRed`, $w_i(\theta) = \frac{1}{n} + \frac{2\beta(f_i(\theta) - \overline{f}(\theta))}{n}$, so the better a client performs, the more it is suppressed by the algorithm. In contrast `Semi-VRed` assigns weights to well-performing clients depending on how bad the worst-off clients perform compared to the average performance (equation 12). As the performance of worst-off clients improves gradually, the algorithm also lets the well-performing ones for further improvement, instead of strictly suppressing them like `VRed`.*

#### 4.3.2 HANDLING EXTREME LABEL SHIFTS

We now provide another interesting interpretation of `Semi-VRed`, related to data heterogeneity in `FL`. In order for an easier interpretability, we assume $P_i(x, y) = P_i(x|y)P_i(y) = P(x|y)P_i(y)$. This means that the class conditional distribution of input $x$ is identical for all clients. Having made this assumption, we define $\overline{\ell}_j(\theta) = \mathbb{E}_{x \sim P(x|y=j)}[\ell(h(x, \theta), j)]$ as the average loss of predictor $h$ on class $j$. We have lemma 3 about the objective function of `Semi-VRed` in equation 9.

**Lemma 3.** *Assuming $P_i(x, y) = P_i(x|y)P_i(y) = P(x|y)P_i(y)$ for $i \in \{1, \dots, n\}$, for any model parameter $\theta$, `Semi-VRed` global objective $F(\theta)$ defined in equation 9 can be expressed as*

$$F(\theta) = \sum_{j=1}^C \overline{P}(j)\overline{\ell}_j(\theta) + \frac{\beta}{n} \sum_{i=1}^n \left( \sum_{j=1}^C [P_i(j) - \overline{P}(j)]\overline{\ell}_j(\theta) \right)_+^2, \tag{13}$$

*where $\overline{P}(j) = \frac{\sum_{i=1}^n P_i(j)}{n}$ is the marginal distribution of class $j$ in the global dataset.*

$P_i(j)$ and $\overline{P}(j)$ show the ratio of class $j$ in the client $i$ local dataset and the global dataset, respectively. Based on equation 13, `Semi-VRed` is capable of improving the performance of the predictor $h$ in extreme class imbalance scenarios: consider when a label $j$ is over-represented in a client $i$'s data (i.e. $P_i(j) \approx 1$) and under-represented in the global data (i.e. $\overline{P}(j) \approx 0$). For a better understanding of how `Semi-VRed` does so, see example 1 in the appendix.

## 5 CONVERGENCE RESULTS: FULL CLIENT PARTICIPATION

In this section, we prove the convergence of our proposed `Semi-VRed` algorithm, when clients fully participate in each round. We make some standard assumptions on the objective functions $f_i$. Specifically, we assume the functions are Lipschitz smooth and strongly convex and also their gradients have bounded norm and local variance:

**Assumption 1** (**smoothness, strong convexity, bounded stochastic gradient and bounded gradient variance**). *Each objective function $f_i$ is $L$-Lipschitz smooth and $\tau$-strongly convex: for any $\theta, \theta' \in \mathbb{R}^d$ and any $i \in [n]$, we have $\|\nabla f_i(\theta) - \nabla f_i(\theta')\| \leq L\|\theta - \theta'\|$ and $f_i(\theta) - \frac{\tau}{2}\|\theta\|^2$ is convex. Also, for any batch $S \sim \mathcal{B}_i^b$ of $b$ i.i.d samples from client $i$ local data, the following inequalities hold (bounded stochastic gradient and bounded local variance):*

$$\mathbb{E}_{S \sim \mathcal{B}_i^b} \left\| \tfrac{1}{|S|} \sum_{(x,y) \in S} \nabla \ell(\theta, (x,y)) \right\|^2 \leq C^2,$$

$$\mathbb{E}_{S \sim \mathcal{B}_i^b} \left\| \tfrac{1}{|S|} \sum_{(x,y) \in S} \nabla \ell(\theta, (x,y)) - \nabla f_i(\theta) \right\|^2 \leq \sigma_{l,i}^2,$$

Note that, according to algorithm 1, each client might have a different number ($K_i$) of mini batches of size $b$, but here we assume $b = 1$ and each client takes $K_i = K$ local steps. Also all clients use the same learning rate $\eta$. In order to prove the convergence of `Semi-VRed`, we also additionally assume boundedness and Lipschitzness for the client losses:

**Assumption 2** ( **boundednes and Lipschitz continuity**). *For any $i \in [n]$, $\theta, \theta' \in \mathbb{R}^d$ and any batch $S \sim \mathcal{B}_i^b$ of $b$ i.i.d. samples, we have:*

$$0 \leq \ell_S(\theta) := \frac{1}{|S|} \sum_{(x,y) \in S} \ell(\theta, (x,y)) \leq \frac{M}{2}$$

$$\|f_i(\theta) - f_i(\theta')\| \leq L_0\|\theta - \theta'\| \tag{14}$$

With the above assumptions, we prove that `Semi-VRed` algorithm converges to the correct solution.

**Theorem 1** (**`Semi-VRed` with full participation**). *Given Assumptions 1 and 2 , let $p_i = \frac{n_i}{N}$, $\nu = \frac{L}{\tau}$ and $\gamma = \max\{8\nu, K\}$. Assume the diminishing learning rate $\eta_t = \frac{2}{\tau(\gamma+t)}$. Then `Semi-VRed` with full participation satisfies:*

$$\mathbb{E}[F(\theta_T)] - F^* \leq \frac{2\nu}{(\gamma + T)} \left( \frac{B}{\tau} + 2(L + BML + 2\beta L_0^2)\|\theta_0 - \theta^*\|^2 \right), \tag{15}$$

*where $F(\theta) = \sum_i p_i \left( f_i(\theta) + \beta \left( f_i(\theta) - \bar{f}(\theta) \right)_+^2 \right)$ and $F^* = \min_\theta F(\theta)$. Also, $B = \sum_{i=1}^n p_i^2(2\sigma_{l,i}^2 + 8\beta^2 M^2 C^2) + 6LF(\theta_0) + 8(K-1)^2 C^2$.*

## 6 EXPERIMENTS

In this section, we evaluate our proposed algorithms for training fair models. From the obtained experimental results, we can observe that `VRed` achieves competitive fairness performance and `Semi-VRed` beats almost all the existing algorithms in terms of multiple fairness metrics. Furthermore, `Semi-VRed` achieves the state of the art performance in terms of the system overall average performance too.

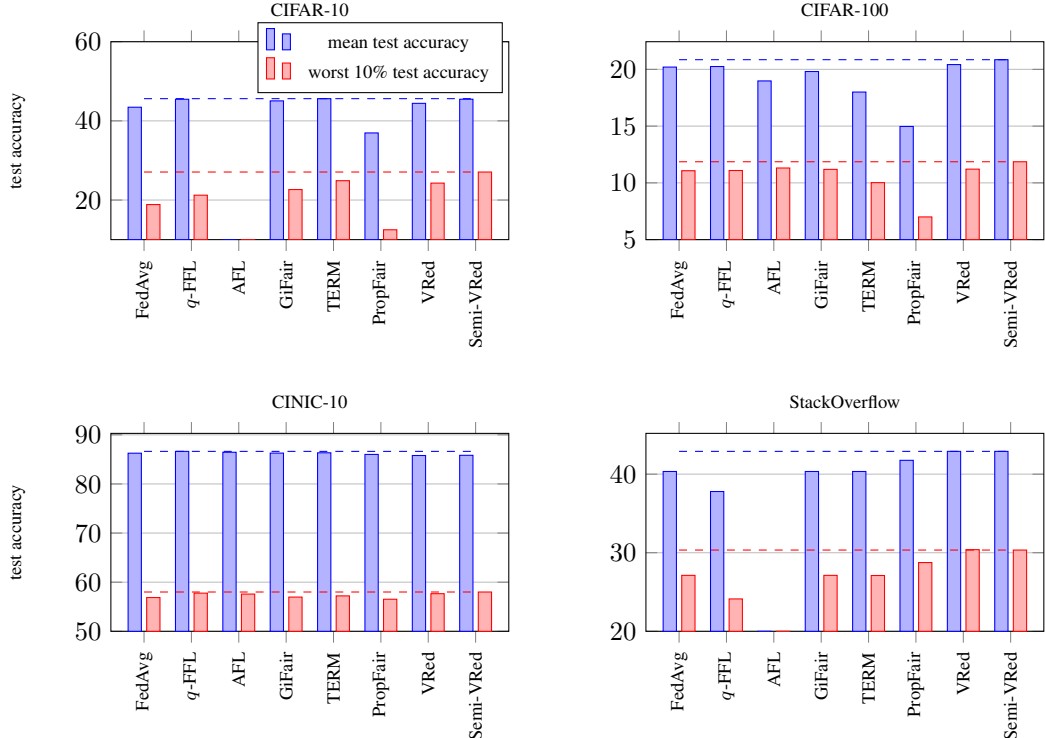

Figure 1: Average and worst 10% test accuracies. **top left:** CIFAR-10, **top right:** CIFAR-100, **bottom left:** CINIC-10, **bottom right:** StackOverflow. Due to divergence, results for AFL on CIFAR-10 and StackOverFlow are not shown. All subfigures share the same legends and axis labels.

## 6.1 EXPERIMENTAL SETUP

In this section, we provide some details about the experiments we conducted to evaluate our algorithms: the details of the datasets, models and their hyperparameters, and the metrics we use to evaluate our algorithms. For detailed explanation of the experiments, see §C in the appendix.

**Datasets** We use four benchmark datasets existing in the literature. The datasets we use include: CIFAR-10/100 (Krizhevsky et al., 2009), CINIC-10 (Darlow et al., 2018) (task of image classification) and StackOverflow (The Tensorflow Federated Authors, 2019) (task of next word prediction). In order to split the data among clients, we use Dirichlet distribution Wang et al. (2019). StackOverflow has a default realistic partition for each client. We follow the same default data distribution.

**Train-Test dataset splitting** After partitioning the dataset among clients, we split the data of each client into train and test sets with a ratio for each dataset. Each client uses its test data to evaluate the common trained model. For more details of the data splittings, see §C in the appendix.

**Models, optimizers and loss functions** For the CIFAR-10/100 and CINIC-10 datasets, we use ResNet-18 He et al. (2016). For the language dataset (StackOverflow), we use LSTMs Hochreiter & Schmidhuber (1997). In order to optimize the models parameters, We use SGD for minimizing average cross entropy loss. For further details, see §C in the appendix.

**Baseline algorithms** We compare our VRed and Semi-VRed algorithms with various fair FL algorithm existing in the literature including: FedAvg McMahan et al. (2017), AFL Mohri et al. (2019), q-FFL Li et al. (2020c), PropFair Zhang et al. (2022a), TERM Li et al. (2020a), GiFair Yue et al. (2021) and Ditto Li et al. (2021).

**Other hyperparameters** We implement an FL setting where different clients participate in all communication rounds with one local epoch at each round. We use 200 communication rounds for all algorithms on the datasets to ensure their convergence. For CIFAR-10/100 and CINIC-10, we partition the data into 50 clients and for language datasets, we partition the data into 20 clients.

Table 1: Comparison between the performance of different algorithms on CIFAR-100. **Second column:** the percentage (%) of suffering clients with improved test accuracy. The value in parentheses shows the amount of test accuracy improvement averaged over suffering clients. **Third column:** the percentage (%) of well-performing clients with degraded test accuracy. The value in parentheses shows the amount of test accuracy improvement averaged over well-performing clients. **Fourth column:** the amount of improvement in the overall mean test accuracy

| Algorithm | Improved suffering clients | Degraded well-performing clients | Overall accuracy improvement |
|---|---|---|---|
| q-FFL | $52.08_{\pm 11.95}$ (+0.69) | $54.21_{\pm 15.38}$ (-0.79) | $+0.04_{\pm 0.41}$ |
| AFL | $51.18_{\pm 9.26}$ (+0.56) | $77.25_{\pm 12.85}$ (-3.50) | $-1.22_{\pm 0.95}$ |
| GiFair | $60.55_{\pm 6.17}$ (+0.86) | $68.22_{\pm 16.13}$ (-2.03) | $-0.40_{\pm 0.61}$ |
| TERM | $23.66_{\pm 5.34}$ (-1.12) | $87.93_{\pm 1.81}$ (-3.57) | $-2.20_{\pm 0.66}$ |
| PropFair | $8.33_{\pm 1.69}$ (-4.05) | $92.41_{\pm 4.28}$ (-6.74) | $-5.23_{\pm 0.96}$ |
| VRed | $60.50_{\pm 12.52}$ (+1.11) | $60.62_{\pm 7.53}$ (-0.94) | $+0.21_{\pm 0.06}$ |
| Semi-VRed | $\mathbf{65.40_{\pm 6.29}}$ **(+1.47)** | $\mathbf{53.17_{\pm 6.75}}$ **(-0.42)** | $\mathbf{+0.64_{\pm 0.30}}$ |

**Evaluation metrics** As we discussed in §4, the ultimate goal of proposing our novel algorithms is to achieve fairness without compromising the system overall average performance. We measure the overall performance with the *mean test accuracy* across clients. In order to measure the fairness in the system, we use the worst 10% test accuracies among clients, which is a standard metric for fairness in FL (Li et al., 2020a;c). In the appendix, we also use other common metrics in the literature for measuring fairness, e.g. the standard deviation of test accuracies (see table 6 in appendix C).

## 6.2 COMPARISON OF VRED AND SEMI-VRED WITH OTHER BASELINE ALGORITHMS

From fig. 1, Semi-VRed outperforms almost all the existing baseline algorithms in terms of the fairness in the system. Furthermore, Semi-VRed improves the system overall average performance (mean test accuracy) for three of the datasets as well. For instance, as can be observed from the results obtained for StackOverflow (see table 6 in § C in the appendix for evaluation in terms of various fairness metrics), Semi-VRed improves both fairness and mean test accuracy by 3% and 2.7%, respectively. Also, we can observe the competitive performance of VRed.

## 6.3 SUPERIORITY OF SEMI-VRED OVER VRED AND THE OTHER BASELINE ALGORITHMS

As discussed in § 2, the existing fair FL algorithms usually suppress the well-performing clients in order to improve the worst-off clients performance. However, Semi-VRed, thanks to its efficient formulation, tries to avoid this. In order to get a better understanding of this, in table 1, we have compared different algorithms based on the amount of performance improvement that they provide: after running the simple vanilla FedAvg on CIFAR-100, we divide the existing 50 clients into two sets: 1. *suffering clients*: those with test accuracies below the FedAvg mean accuracy (22 clients in our experiment) 2. *well-performing clients*: those with test accuracies above the FedAvg mean accuracy (28 clients). Then, we run each of the other algorithms and compare their performance improvement with each other. The results clearly delivers two important messages: 1. the existing algorithms either more or less suppress the well-performing clients or cannot improve them, due to the more attention that they pay to the worst-off clients 2. Semi-VRed has the least suppression of well-performing clients (53.17% with a small average degradation of -0.42), and the highest average improvement of suffering clients (65.40% with an average accuracy improvement of +1.47), which results in improving both the fairness and the system overall average performance simultaneously.

## 7 CONCLUSION

In this work, we introduced two novel fair FL algorithms: VRed and Semi-VRed. In order to resolve the drawback of most of the existing fair FL algorithms, which is suppression of well-performing clients, we propose Semi-VRed, which uses an efficient method for measuring performance inequality in a FL system. Our experimental results show that Semi-VRed not only improves the worst-off clients performance, but also improves the system overall average performance as well. Accordingly, Semi-VRed achieves SoTA performance in terms of both the overall average accuracy and fairness, measured in terms of various common fairness metrics.

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

# Appendix for *Semi-Variance Reduction for Fair Federated Learning*

## A   PROOFS

**Lemma 1.** *For any model parameter $\theta$, the gradient of the global objective $F(\theta)$ defined in equation 3 can be expressed as*

$$\nabla F(\theta) = \sum_{i=1}^{n} w_i(\theta) \nabla f_i(\theta), \;\; w_i(\theta) = \frac{1}{n} + \frac{2\beta(f_i(\theta) - \overline{f}(\theta))}{n}, \;\; \overline{f}(\theta) = \frac{\sum_i f_i(\theta)}{n}. \tag{8}$$

*Proof.* From equation 3 and with $p_i = \frac{1}{n}$, we have:

$$\begin{aligned}
n\nabla F(\theta) &= \sum_i \nabla f_i(\theta) + 2\beta \sum_i \left[ \left( f_i(\theta) - \overline{f}(\theta) \right) \left( \nabla f_i(\theta) - \nabla \overline{f}(\theta) \right) \right] \\
&= \sum_i \nabla f_i(\theta) + 2\beta \sum_i \left[ \left( f_i(\theta) - \overline{f}(\theta) \right) \nabla f_i(\theta) - \left( f_i(\theta) - \overline{f}(\theta) \right) \nabla \overline{f}(\theta) \right] \\
&= \sum_i \left( 1 + 2\beta(f_i(\theta) - \overline{f}(\theta)) \right) \nabla f_i(\theta) - 2\beta \sum_i \left( f_i(\theta) - \overline{f}(\theta) \right) \nabla \overline{f}(\theta) \\
&= \sum_i \left( 1 + 2\beta(f_i(\theta) - \overline{f}(\theta)) \right) \nabla f_i(\theta)
\end{aligned} \tag{16}$$

Hence,

$$\nabla F(\theta) = \sum_i \frac{1 + 2\beta(f_i(\theta) - \overline{f}(\theta))}{n} \nabla f_i(\theta) \tag{17}$$

$\square$

**Derivation of equation 6**

$$\begin{aligned}
\sum_i f_i + \beta \sum_i \left| f_i(\theta) - \frac{1}{n} \sum_j f_j(\theta) \right|^2 &= \sum_i f_i + \beta \sum_i \left| \frac{n-1}{n} f_i(\theta) - \frac{1}{n} \sum_{j \neq i} f_j(\theta) \right|^2 \\
&= \sum_i f_i + \frac{\beta}{n^2} \sum_i \left| \sum_{j \neq i} (f_i(\theta) - f_j(\theta)) \right|^2 \\
&\leq \sum_i f_i + \frac{\beta}{n^2} \sum_i \sum_{j \neq i} \left| f_i(\theta) - f_j(\theta) \right|^2 \\
&= \sum_i f_i + \frac{2\beta}{n^2} \sum_{j \neq i} \left| f_i(\theta) - f_j(\theta) \right|^2.
\end{aligned} \tag{18}$$

**Lemma 2.** *In each communication round between the clients and the server, let $>_C$ denote the set of clients whose local loss function is greater than the average loss function $\overline{f}(\theta)$. For any model parameter $\theta$, the gradient of the global objective $F(\theta)$ defined in equation 9 can be expressed as*

$$\nabla F(\theta) = \sum_{i=1}^{n} w_i(\theta) \nabla f_i(\theta), \tag{11}$$

*where:*

$$\overline{f}(\theta) = \frac{\sum_i f_i(\theta)}{n}, \quad w_i(\theta) = \begin{cases} \frac{1}{n} + \frac{2\beta(f_i(\theta) - \overline{f}(\theta))}{n} - \frac{2\beta \sum_{j>_c}(f_j(\theta) - \overline{f}(\theta))}{n^2}, & \text{if } i \in >_c \\ \frac{1}{n} - \frac{2\beta \sum_{j>_c}(f_j(\theta) - \overline{f}(\theta))}{n^2}, & \text{if } i \notin >_c \end{cases}$$

(12)

*Proof.* From equation 9 and with $p_i = \frac{1}{n}$, we have:

$$\begin{aligned} n\nabla F(\theta) &= \sum_i \nabla f_i(\theta) + 2\beta \sum_{i \in >_c} \left[ \Big( f_i(\theta) - \overline{f}(\theta) \Big) \Big( \nabla f_i(\theta) - \nabla \overline{f}(\theta) \Big) \right] \\ &= \sum_i \nabla f_i(\theta) + 2\beta \sum_{i \in >_c} \left[ \Big( f_i(\theta) - \overline{f}(\theta) \Big) \nabla f_i(\theta) - \Big( f_i(\theta) - \overline{f}(\theta) \Big) \nabla \overline{f}(\theta) \right] \\ &= \sum_{i \notin >_c} \nabla f_i(\theta) + \sum_{i \in >_c} \Big( 1 + 2\beta(f_i(\theta) - \overline{f}(\theta)) \Big) \nabla f_i(\theta) - 2\beta \sum_{i \in >_c} \Big( f_i(\theta) - \overline{f}(\theta) \Big) \nabla \overline{f}(\theta) \end{aligned}$$

(19)

The last term in the above equation can be written as:

$$\begin{aligned} &- 2\beta \sum_{i \in >_c} \Big( f_i(\theta) - \overline{f}(\theta) \Big) \nabla \overline{f}(\theta) \\ &= -\left[ \frac{2\beta}{n} \Big( \sum_{i \in >_c} f_i(\theta) \Big) \times \Big( \sum_j \nabla f_j(\theta) \Big) \right] + \left[ \frac{2\beta}{n} \Big( \sum_{i \in >_c} \overline{f}(\theta) \Big) \times \Big( \sum_j \nabla f_j(\theta) \Big) \right] \\ &= -\left[ \frac{2\beta}{n} \Big( \sum_{i \in >_c} f_i(\theta) - \overline{f}(\theta) \Big) \times \Big( \sum_j \nabla f_j(\theta) \Big) \right] \end{aligned}$$

(20)

Hence,

$$\begin{aligned} n\nabla F(\theta) &= \sum_{i \in >_c} \Big( 1 + 2\beta(f_i(\theta) - \overline{f}(\theta)) - \frac{2\beta}{n} \Big( \sum_{j>_c} f_j(\theta) - \overline{f}(\theta) \Big) \Big) \nabla f_i(\theta) \\ &+ \sum_{i \notin >_c} \Big( 1 - \frac{2\beta}{n} \Big( \sum_{j \in >_c} f_j(\theta) - \overline{f}(\theta) \Big) \Big) \nabla f_i(\theta) \end{aligned}$$

(21)

Therefore,

$$\begin{aligned} \nabla F(\theta) &= \sum_{i \in >_c} \Big( \frac{1 + 2\beta(f_i(\theta) - \overline{f}(\theta)) - \frac{2\beta}{n} \Big( \sum_{j \in >_c} f_j(\theta) - \overline{f}(\theta) \Big)}{n} \Big) \nabla f_i(\theta) \\ &+ \sum_{i \notin >_c} \Big( \frac{1 - \frac{2\beta}{n} \Big( \sum_{j \in >_c} f_j(\theta) - \overline{f}(\theta) \Big)}{n} \Big) \nabla f_i(\theta) \end{aligned}$$

(22)

$\square$

**Lemma 3.** *Assuming $P_i(x,y) = P_i(x|y)P_i(y) = P(x|y)P_i(y)$ for $i \in \{1,\ldots,n\}$, for any model parameter $\theta$,* Semi-VRed *global objective $F(\theta)$ defined in equation 9 can be expressed as*

$$F(\theta) = \sum_{j=1}^C \overline{P}(j)\overline{\ell}_j(\theta) + \frac{\beta}{n} \sum_{i=1}^n \Big( \sum_{j=1}^C [P_i(j) - \overline{P}(j)]\overline{\ell}_j(\theta) \Big)_+^2,$$

(13)

*where $\overline{P}(j) = \frac{\sum_{i=1}^n P_i(j)}{n}$ is the marginal distribution of class $j$ in the global dataset.*

*Proof.* From equation 9 and with $p_i = \frac{1}{n}$, we have:

$$
\begin{aligned}
\overline{f}(\theta) = \sum_{i=1}^{n} \frac{f_i(\theta)}{n} &= \frac{1}{n} \sum_{i=1}^{n} \Big[ \mathbb{E}_{(x,y) \sim p_i(x,y)}[\ell(h(x,\theta), y)] \Big] \\
&= \frac{1}{n} \sum_{i=1}^{n} \Big[ \sum_{j=1}^{C} p_i(j) \times \mathbb{E}_{(x,y) \sim p(x|y=j)}[\ell(h(x,\theta), j)] \Big] \\
&= \frac{1}{n} \sum_{i=1}^{n} \Big[ \sum_{j=1}^{C} p_i(j) \overline{\ell}_j(\theta)] \Big] \\
&= \sum_{j=1}^{C} \Big[ \big( \frac{\sum_{i=1}^{n} p_i(j)}{n} \big) \overline{\ell}_j(\theta) \Big] \\
&= \sum_{j=1}^{C} \overline{p}(j) \overline{\ell}_j(\theta),
\end{aligned}
\tag{23}
$$

where $\overline{p}(j) = \frac{\sum_{i=1}^{n} p_i(j)}{n}$ is the ratio of data points with label $j$ in the global dataset. Similarly, we have:

$$
f_i(\theta) = \sum_{j=1}^{C} p_i(j) \overline{\ell}_j(\theta).
\tag{24}
$$

By plugging in the above equivalences for $f_i(\theta)$ and $\overline{f}(\theta)$ into equation 9, we get to equation 13. $\quad\square$

Due to the similarity of `Semi-VRed`'s objective function to that of `FedAvg`, we build its convergence proof on top of the convergence proof for `FedAvg` in Li et al. (2020d). We refer the reader to the work for the detailed proof.

We now prove the convergence of our `Semi-VRed` algorithm.

**Theorem 1** (**Semi-VRed with full participation**). *Given Assumptions 1 and 2 , let $p_i = \frac{n_i}{N}$, $\nu = \frac{L}{\tau}$ and $\gamma = \max\{8\nu, K\}$. Assume the diminishing learning rate $\eta_t = \frac{2}{\tau(\gamma + t)}$. Then `Semi-VRed` with full participation satisfies:*

$$
\mathbb{E}[F(\theta_T)] - F^* \leq \frac{2\nu}{(\gamma + T)} \left( \frac{B}{\tau} + 2(L + BML + 2\beta L_0^2) \|\theta_0 - \theta^*\|^2 \right),
\tag{15}
$$

*where $F(\theta) = \sum_i p_i \Big( f_i(\theta) + \beta \big( f_i(\theta) - \bar{f}(\theta) \big)_+^2 \Big)$ and $F^* = \min_\theta F(\theta)$. Also, $B = \sum_{i=1}^{n} p_i^2 (2\sigma_{l,i}^2 + 8\beta^2 M^2 C^2) + 6LF(\theta_0) + 8(K-1)^2 C^2$.*

*Proof.* We first rewrite a simplified version of the `Semi-VRed` objective function (equation 9) in the following.

$$
F(\theta) = \sum_i p_i G_i(\theta) = \sum_i p_i \big( f_i(\theta) + \beta \left( f_i(\theta) - \mu \right)_+^2 \big),
\tag{25}
$$

where $G_i(\theta) = f_i(\theta) + \beta \left( f_i(\theta) - \mu \right)_+^2$, and also, $\mu = \Sigma_i p_i f_i$ is fixed during clients local computations. In the beginning of each communication round, we update $\mu$ for the next round of local computations. In other words:

$$
\mu_{t+1} = \sum_{i=1}^{n} p_i f_i(\theta_{t+1}).
\tag{26}
$$

With these notations, it suffices to find the constants in Assumption 1 for $G_i(\theta)$.

**Smoothness** We have:

$$G_i(\theta) = f_i(\theta) + \beta(f_i(\theta) - \mu)_+^2 = \begin{cases} f_i(\theta) & , \text{if } 0 \leq f_i(\theta) \leq \mu \\ f_i(\theta) + \beta(f_i(\theta) - \mu)^2 & , \text{if } f_i(\theta) > \mu \end{cases} \tag{27}$$

In the first case, $G_i$ is smooth with the same smoothness parameter of $f_i(\theta)$:

$$\|\nabla G_i(\theta) - \nabla G_i(\theta')\| \leq L\|\theta - \theta'\|. \tag{28}$$

In the second case, we have:

$$\nabla^2 G_i(\theta) = \nabla^2 f_i(\theta) + 2\beta((f_i(\theta) - \mu)\nabla^2 f_i(\theta) + \nabla f_i(\theta)\nabla f_i(\theta)^\top)$$

$$\preceq L + 2\beta(\frac{M}{2}\nabla^2 f_i(\theta) + \nabla f_i(\theta)\nabla f_i(\theta)^\top)$$

$$\preceq (L + \beta M L + 2\beta L_0^2)I, \tag{29}$$

where in the last line, we used Assumption 2 and the following:

$$\|\nabla f_i(\theta)\nabla f_i(\theta)^\top\|_{sp} = \sup_{\|u\|=1} \sup_{\|v\|=1} \langle \nabla f_i(\theta)\nabla f_i(\theta)^\top u; v\rangle$$

$$= \sup_{\|u\|=1} \sup_{\|v\|=1} (\nabla f_i(\theta)^\top u)^\top \nabla f_i(\theta)^\top v = \|\nabla f_i(\theta)\|^2 \leq L_0^2, \tag{30}$$

where in the second line we used Cauchy–Schwarz inequality and Assumption 2. Hence, from eq. (28) and eq. (29), we conclude that $G_i(\theta)$ is Lipschitz smooth:

$$\|\nabla G_i(\theta) - \nabla G_i(\theta')\| \leq (L + \beta M L + 2\beta L_0^2)\|\theta - \theta'\|. \tag{31}$$

**Strong Convexity** From, eq. (27), we have:

$$\nabla^2 G_i(\theta) = \begin{cases} \nabla^2 f_i(\theta) & , \text{if } 0 \leq f_i(\theta) \leq \mu \\ \nabla^2 f_i(\theta) + 2\beta(f_i(\theta) - \mu)\nabla^2 f_i(\theta) + 2\beta\nabla f_i(\theta)\nabla f_i(\theta)^\top & , \text{if } f_i(\theta) > \mu \end{cases} \tag{32}$$

From the above derivation of $\nabla^2 G_i(\theta)$ and that $f_i(\theta)$ is $\tau$-strongly convex, we can immediately conclude that $G_i(\theta)$ is also $\tau$-strongly convex.

**local gradient variance constants** For the local variance term, we define $\varphi(t) = t + \beta(t - \mu)_+^2$. We have:

$$\|\nabla G_i(\theta) - \nabla(\varphi \circ \ell_S)(\theta)\|$$

$$= \left\| \Big(\nabla f_i(\theta) + 2\beta(f_i(\theta) - \mu)_+ \nabla f_i(\theta)\Big) - \Big(\nabla \ell_S(\theta) + 2\beta(\ell_S(\theta) - \mu)_+ \nabla \ell_S(\theta)\Big) \right\|$$

$$\leq \|\nabla f_i(\theta) - \nabla \ell_S(\theta)\| + 2\beta\|(f_i(\theta) - \mu)_+ \nabla f_i(\theta)\| + 2\beta\|\ell_S(\theta) - \mu)_+ \nabla \ell_S(\theta)\|$$

$$\leq \|\nabla f_i(\theta) - \nabla \ell_S(\theta)\| + \beta M\|\nabla f_i(\theta)\| + \beta M\|\nabla \ell_S(\theta)\|$$

$$\leq \|\nabla f_i(\theta) - \nabla \ell_S(\theta)\| + 2\beta M C \tag{33}$$

where in line four, we used Assumption 2 and in line five, we used 1. By taking the square on both sides and the expectation over $S \sim \mathcal{B}_i^b$, we get:

$$\mathbb{E}_{S\sim\mathcal{B}_i^b}\|\nabla G_i(\theta) - \nabla(\varphi \circ \ell_S)(\theta)\|^2 \leq \mathbb{E}_{S\sim\mathcal{B}_i^b}\Big(\|\nabla f_i(\theta) - \nabla \ell_S(\theta)\| + 2\beta M C\Big)^2$$

$$\leq \mathbb{E}_{S\sim\mathcal{B}_i^b}\Big(2\|\nabla f_i(\theta) - \nabla \ell_S(\theta)\|^2 + 8\beta^2 M^2 C^2\Big)$$

$$= \Big(2\sigma_{l,i}^2 + 8\beta^2 M^2 C^2\Big). \tag{34}$$

In the third line, we used $(a + b)^2 \leq 2(a^2 + b^2)$. We also used Assumption 1 in the same line.

$\square$

## B  EXAMPLES

We borrow the following example on class imbalance in `FL` from Shen et al. (2022) to provide a better understanding of lemma 3. The following example shows an extreme class imbalance, which `Semi-VRed` can handle efficiently.

**Example 1.** *Let $u$ be the uniform distribution over the existing $C$ classes. Also, let $\delta_c$ be the Dirac distribution of class c. Now, without loss of generality, lets assume that $C = 2$ (binary classification problem). For the $n$ existing clients, we have:*

$$p_i(y) = \begin{cases} \alpha u + (1 - \alpha)\delta_1 & \textit{if } i = 1 \\ \alpha u + (1 - \alpha)\delta_2 & \textit{if } i \in \{2, \ldots, n\} \end{cases} \tag{35}$$

*Accordingly, we have:*

$$p_i(1) = \begin{cases} 1 - \dfrac{\alpha}{2} & \textit{if } i = 1 \\ \dfrac{\alpha}{2} & \textit{if } i \in \{2, \ldots, n\} \end{cases} \tag{36}$$

$$p_i(2) = \begin{cases} \dfrac{\alpha}{2} & \textit{if } i = 1 \\ 1 - \dfrac{\alpha}{2} & \textit{if } i \in \{2, \ldots, n\} \end{cases} \tag{37}$$

*Therefore,*

$$f_i(\theta) = \begin{cases} (1 - \dfrac{\alpha}{2})\bar{\ell}_1(\theta) + \dfrac{\alpha}{2}\bar{\ell}_2(\theta) & \textit{if } i = 1 \\ \dfrac{\alpha}{2}\bar{\ell}_1(\theta) + (1 - \dfrac{\alpha}{2})\bar{\ell}_2(\theta) & \textit{if } i \in \{2, \ldots, n\} \end{cases} \tag{38}$$

*Hence,*

$$\bar{f}(\theta) = \left(\frac{\alpha}{2} + \frac{1 - \alpha}{n}\right)\bar{\ell}_1(\theta) + \left(\frac{\alpha}{2} + \frac{(1 - \alpha)(n - 1)}{n}\right)\bar{\ell}_2(\theta) \tag{39}$$

*Clearly, we can see that if $\alpha \approx 0$ and $n$ is large, then $\bar{\ell}_1(\theta)$, which is the loss over the minority data will have a small weight, which leads to $\bar{\ell}_1(\theta)$ being larger than $\bar{\ell}_2(\theta)$ and poor performance on the minority class 1. Now, if we rewrite the `Semi-VRed` objective function (equation 9), we have:*

$$F(\theta) = \left(\frac{\alpha}{2} + \frac{1 - \alpha}{n}\right)\bar{\ell}_1(\theta) + \left(\frac{\alpha}{2} + \frac{(1 - \alpha)(n - 1)}{n}\right)\bar{\ell}_2(\theta) + \frac{\beta(n - 1)^2(1 - \alpha)^2}{n^3}\left(\bar{\ell}_1(\theta) - \bar{\ell}_2(\theta)\right)^2 \tag{40}$$

*For $\alpha \approx 0$:*

$$F(\theta) \approx \bar{\ell}_2(\theta) + \frac{\beta}{n}\left(\bar{\ell}_1(\theta) - \bar{\ell}_2(\theta)\right)^2, \tag{41}$$

*which improves $\bar{\ell}_2(\theta)$, thanks to its regularization term. Hence, the performance of client 1 and consequently, fairness in the system will improve.*

## C    EXPERIMENTAL SETUP

In this section, we provide more experimental details that are deferred from the main paper. For each experiment, we report the average result obtained from three runs with different random seeds. For our experiments, we used an internal GPU server with six NVIDIA Tesla P100. The experiments last about 4 weeks in total.

### C.1    DATASETS AND MODELS

In this subsection, we describe the datasets we use in our experiments. For all the datasets we use a batch size of 64.

**CIFAR-10/100** (Krizhevsky et al., 2009) are two image classification datasets vastly used in the literature as benchmark datasets. Each of these datasests contains 50000 sample images with 10/100 balanced classes for CIFAR-10 and CIFAR-100, respectively. We use Dirichlet allocation (Wang et al., 2019) to distribute the data among 50 clients with label shift: we split the set of samples from class $k$ ($\mathcal{S}_k$) to $n$ subsets ($\mathcal{S}_k = \mathcal{S}_{k,1} \cup \mathcal{S}_{k,2} \cup \ldots \cup \mathcal{S}_{k,n}$), where $n$ is the number of clients and $\mathcal{S}_{k,j}$ corresponds to the client $j$. We do the split based on Dirichlet distribution with parameter 0.05 (`Dir(0.05)`). When the split is done for all classes, we gather the samples corresponding to each client from different classes: assuming there are $C$ classes in total $\mathcal{S}_{1,j} \cup \mathcal{S}_{2,j} \cup \ldots \cup \mathcal{S}_{C,j}$ is the data allocated to the client $j$. Having allocated the data of each client, we split them into train and test set for each client. The train-test split ratio is 50-50 and 60-40 for CIFAR-10 and CIFAR-100, respectively.

**CINIC-10** (Darlow et al., 2018) is another benchmark vision dataset that we use in our experiments. There are a total of 270,000 sample images, which we distribute with label shift between 50 clients based on `Dir(0.5)` distribution Wang et al. (2019). We then randomly split the data of each client into train and test sets with split ratio 50-50.

**StackOverflow** (The Tensorflow Federated Authors, 2019) is a language dataset consisting of Shakespeare dialogues for the task of next word prediction. There is a natural heterogeneous partition of the dataset and we treat each speaking role as a client. We filter out the clients (speaking roles) with less than 10,000 samples from the original dataset and randomly select 20 clients from the remaining. Finally, we split the data of each client into train and test sets with a ratio of 50-50.

Table 2 provides a summary of the datasets we used and the models used for each of them.

Table 2: Details of the experiments and the datasets. ResNet-18: residual neural network (He et al., 2016). GN: Group Normalization (Wu & He, 2018); RNN: Recurrent Neural Network; LSTM: Long Short-Term Memory layer; FC: fully connected layer.

| Dataset | Train samples | Test samples | Partition method | clients | Model |
|---------|---------------|--------------|------------------|---------|-------|
| CIFAR-10 | 24959 | 25041 | `Dir(0.05)` | 50 | ResNet-18 + GN |
| CIFAR-100 | 39445 | 10555 | `Dir(0.05)` | 50 | ResNet-18 + GN |
| CINIC-10 | 134713 | 134966 | `Dir(0.5)` | 50 | ResNet-18 + GN |
| StackOverflow | 109671 | 109621 | realistic partition | 20 | RNN (1 LSTM + 2 FC) |

### C.2    ALGORITHMS AND THEIR HYPERPARAMETERS

We use most recent fair `FL` algorithms existing in the literature as our baseline algorithms, including: `FedAvg` (McMahan et al., 2017), `q-FFL` (Li et al., 2020c), `AFL` (Mohri et al., 2019), `PropFair` (Zhang et al., 2022a), `TERM` (Li et al., 2020a), `GiFair` (Yue et al., 2021). For each pair of dataset and algorithm, we find the best learning rate based on a grid search. In the following, we have reported the learning rate grid we use for each dataset:

- CIFAR-10: {`1e-3`, `2e-3`, `5e-3`, `1e-2`, `2e-2`, `5e-2`};
- CIFAR-100: {`1e-3`, `2e-3`, `5e-3`, `1e-2`, `2e-2`, `5e-2`};
- CINIC-10: {`1e-3`, `2e-3`, `5e-3`, `1e-2`, `2e-2`, `5e-2`};

Table 3: Details of the existing fairfl algorithms. $f_i$ is the loss function of the client $i$.

| FL algorithm | Objective | Reference |
|---|---|---|
| FedAvg | $\sum_i f_i$ | McMahan et al. (2017) |
| AFL | $\max_i f_i$ | Mohri et al. (2019) |
| q-FFL | $\sum_i f_i^{q+1}$ | Li et al. (2020c) |
| TERM | $\sum_i e^{\alpha f_i}$ | Li et al. (2020a) |
| PropFair | $-\sum_i \log(M - f_i)$ | Zhang et al. (2022a) |
| GiFair | $\sum_i f_i + \lambda \sum_{i \neq j} \lvert f_i - f_j \rvert$ | Yue et al. (2021) |
| VRed | $\sum_i f_i + \beta \sum_i \left( f_i(\theta) - \frac{1}{n} \sum_j f_j(\theta) \right)^2$ | this work |
| Semi-VRed | $\sum_i f_i + \beta \sum_i \left( f_i(\theta) - \frac{1}{n} \sum_j f_j(\theta) \right)_+^2$ | this work |

- StackOverflow: $\{$`1e-2, 5e-2, 1e-1, 5e-1, 1`$\}$.

The best learning rate used for each (dataset, algorithm) pair is reported in Table 4.

Table 4: The best learning rates used for training each algorithm on different datasets.

| Datasets | FedAvg | q-FFL | AFL | TERM | PropFair | GiFair | VRed | Semi-VRed |
|---|---|---|---|---|---|---|---|---|
| CIFAR-10 | 5e-3 | 5e-3 | 5e-3 | 1e-2 | 1e-2 | 5e-3 | 5e-3 | 5e-3 |
| CIFAR-100 | 2e-3 | 2e-3 | 5e-3 | 1e-2 | 1e-2 | 5e-3 | 5e-3 | 5e-3 |
| CINIC-10 | 1e-2 | 5e-3 | 1e-2 | 1e-2 | 2e-2 | 2e-2 | 5e-3 | 5e-3 |
| StackOverflow | 2e-1 | 5e-2 | 5e-2 | 2e-1 | 5e-1 | 2e-1 | 5e-1 | 5e-1 |

We now explain the algorithms we use and how we tune their hyperparameters. We adapt TERM with only client-level fairness ($\alpha > 0$) and no sample-level fairness ($\tau = 0$). We tune the hyperparameter $\alpha$ for each dataset based on a grid search in the grid $\{0.01, 0.1, 0.5, 1\}$. We have reported the best value of $\alpha$ for each dataset in Table 5. For AFL, there are two hyperparameters: $\gamma_w$ and $\gamma_\lambda$. We tune the learning rate $\gamma_w$ from the corresponding grid and choose the default value $\gamma_\lambda = 0.1$. For $q$-FFL, we use the $q$-FedAvg algorithm (Li et al., 2020c). We also tune the hyperparameter $q$ from the grid $\{0.01, 0.1, 1\}$. We find that for all the used datasets, $q = 0.1$ has the best peformance (as reported in Table 5). We also tried larger values out of the grid and they often lead to divergence of the q-FFL algorithm. We adopt the Global GiFair model (Yue et al., 2021), which results in a single global model. In order to have client-level fairness, we treat each client as a group of size 1. For tuning the regularization weight of GiFair ($\lambda$), we follow (Yue et al., 2021). As stated in the paper, there is an upper-bound for $\lambda$ (see §3 in the paper). For our experiments, the upper-bound is $\lambda \leq \min_i\{\frac{w_i}{n-1}\}$, where $w_i$ is the ratio of total samples allocated to the client $i$ and $n$ is the number of clients. We try four different values in the interval and choose the best one. When the number of clients is large, this upper-bound is small, and for all of our datasets, this upper-bound was the best value, as reported in Table 5. We tune $M$ for the PropFair algorithm based on a grid search in $\{2, 3, 4, 5\}$. Finally, for our VRed and Semi-VRed algorithms, we tune the regularization weight $\beta$ based on grid search on the grid $\{0.01, 0.05, 0.1, 0.2, 0.5, 1\}$. Larger values of $\beta$ often resulted in the divergence of the algorithms. We have reported the best value of all of the hyperparameters for each dataset in Table 5.

## C.3 Detailed results

In Table 6, we report detailed results obtained from the algorithms we study in this work. We use a default batch size of 64 for all the experiments. The statistics we report include: 1. the average test accuracy across clients (overall average performance) 2. the standard deviation of test accuracies across clients 3. the lowest (worst) test accuracy among clients 4. the lowest 10% test accuracies 5. the lowest 20% test accuracies 6. the highest 10% test accuracies. For each experiment, we report the average result obtained from three runs with different random seeds. As can be observed, our proposed algorithms VRed and Semi-VRed consistently beat almost all the baseline algorithms

Table 5: The best values of hyperparameters used for different datasets, chosen based on grid search.

| Algorithm | CIFAR-10 | CIFAR-100 | CINIC-10 | StackOverflow |
|---|---|---|---|---|
| **q-FFL** $q$ | 1e-1 | 1e-1 | 1e-1 | 1e-1 |
| **TERM** $\alpha$ | 1e-2 | 5e-1 | 5e-1 | 5e-1 |
| **GiFair** $\lambda$ | 6e-5 | 2.6e-4 | 5e-5 | 2.4e-3 |
| **PropFair** $M$ | 3 | 3 | 5 | 4 |
| **VRed** $\beta$ | 5e-1 | 1e-1 | 2e-1 | 1e-1 |
| **Semi-VRed** $\beta$ | 5e-1 | 1e-2 | 2e-1 | 2e-1 |

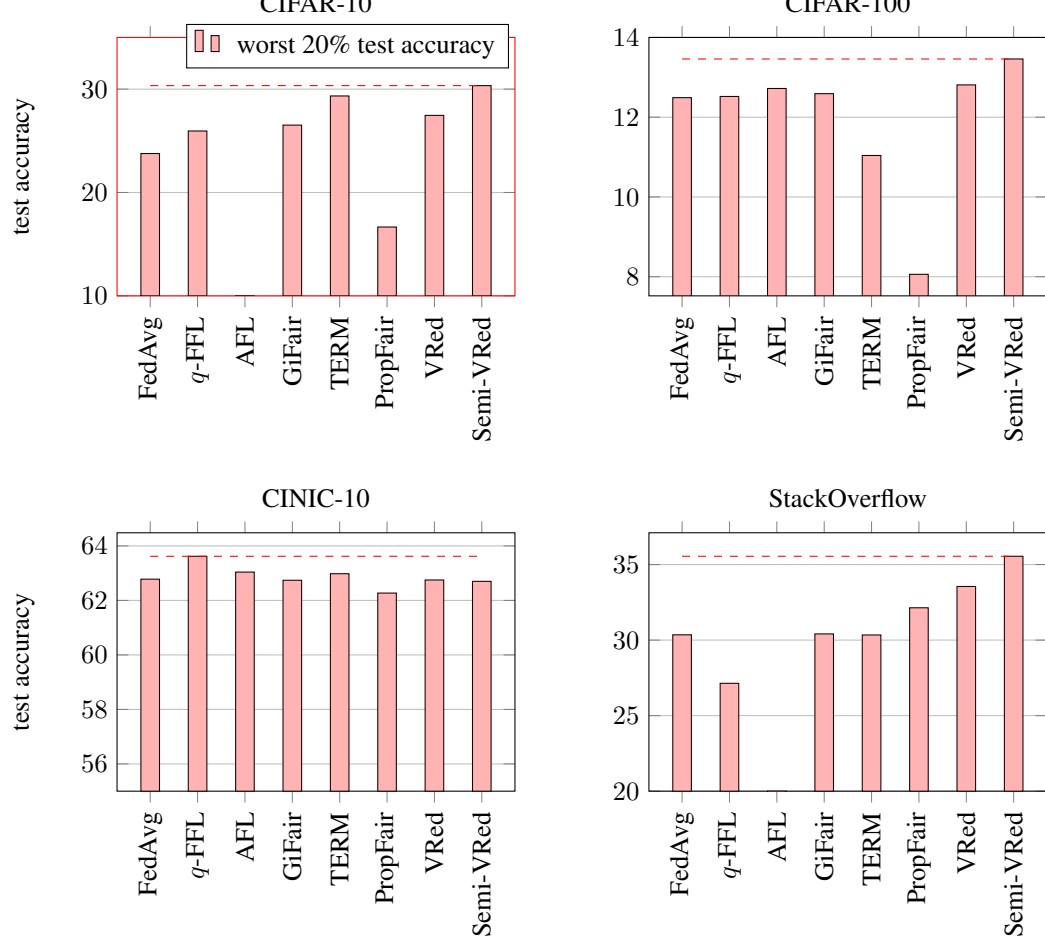

Figure 2: Worst 20% test accuracies for different algorithms. **top left:** CIFAR-10, **top right:** CIFAR-100, **bottom left:** CINIC-10, **bottom right:** StackOverflow. Due to divergence, results for AFL on CIFAR-10 and StackOverFlow are not shown. All subfigures share the same legends and axis labels.

across different datasets in terms of various fairness metrics. Also, Semi-VRed can improve the overall average performance (mean test accuracy) for three of the datsets as well.

Following Figure 1, we have compared our proposed algorithms with the baseline algorithms in terms of their worst 20% test accuracies as well and the visualized results are shown in Figure 2.

Table 6: Comparison among federated learning algorithms on CIFAR-10, CIFAR-100, CINIC-10 and StackOverflow datasets with test accuracies (%) from clients. All algorithms are fine-tuned. **Mean**: the average test accuracy across all clients; **Std**: standard deviation of clients test accuracies; **Worst**: the worst test accuracy among clients; **Worst (10/20%)**: the worst 10/20% test accuracies of clients; **Best (10%)**: the best 10% test accuracies of clients.

| Dataset | Algorithm | Mean | Std | Worst | Worst (10%) | Worst (20%) | Best (10%) |
|---|---|---|---|---|---|---|---|
| CIFAR-10 | FedAvg, Ditto | $43.45_{\pm0.60}$ | $14.33_{\pm0.62}$ | $9.35_{\pm3.13}$ | $18.86_{\pm0.99}$ | $23.77_{\pm0.70}$ | $68.97_{\pm0.81}$ |
| | $q$-FFL | $45.46_{\pm0.74}$ | $14.31_{\pm2.03}$ | $18.71_{\pm3.36}$ | $21.23_{\pm3.06}$ | $25.95_{\pm3.51}$ | $72.31_{\pm2.88}$ |
| | AFL | - | - | - | - | - | - |
| | GiFair | $45.05_{\pm0.64}$ | $12.93_{\pm0.44}$ | $16.79_{\pm3.55}$ | $22.65_{\pm2.03}$ | $26.52_{\pm0.76}$ | $65.62_{\pm2.59}$ |
| | TERM | $\mathbf{45.61}_{\pm1.03}$ | $\mathbf{12.24}_{\pm0.56}$ | $13.80_{\pm5.25}$ | $24.89_{\pm1.37}$ | $29.34_{\pm0.61}$ | $68.65_{\pm1.27}$ |
| | PropFair | $36.95_{\pm0.21}$ | $15.16_{\pm1.33}$ | $1.14_{\pm1.62}$ | $12.49_{\pm0.28}$ | $16.66_{\pm1.31}$ | $66.04_{\pm4.24}$ |
| | VRed | $44.43_{\pm0.88}$ | $13.05_{\pm1.32}$ | $18.61_{\pm3.12}$ | $24.28_{\pm2.22}$ | $27.46_{\pm1.56}$ | $69.31_{\pm3.48}$ |
| | Semi-VRed | $45.47_{\pm0.10}$ | $12.58_{\pm0.23}$ | $\mathbf{19.04}_{\pm6.73}$ | $\mathbf{27.08}_{\pm1.76}$ | $\mathbf{30.34}_{\pm1.05}$ | $\mathbf{72.50}_{\pm0.88}$ |
| CIFAR-100 | FedAvg, Ditto | $20.20_{\pm0.31}$ | $6.50_{\pm0.21}$ | $\mathbf{10.36}_{\pm0.69}$ | $11.07_{\pm0.54}$ | $12.49_{\pm0.51}$ | $33.88_{\pm0.09}$ |
| | $q$-FFL | $20.25_{\pm0.11}$ | $6.30_{\pm0.27}$ | $9.66_{\pm0.33}$ | $11.09_{\pm0.67}$ | $12.52_{\pm0.46}$ | $33.96_{\pm0.90}$ |
| | AFL | $18.98_{\pm0.71}$ | $\mathbf{4.91}_{\pm0.37}$ | $9.81_{\pm0.69}$ | $11.31_{\pm0.18}$ | $12.72_{\pm0.21}$ | $28.68_{\pm1.71}$ |
| | GiFair | $19.81_{\pm0.32}$ | $5.74_{\pm0.16}$ | $9.35_{\pm0.34}$ | $11.19_{\pm0.24}$ | $12.59_{\pm0.49}$ | $32.30_{\pm0.32}$ |
| | TERM | $18.00_{\pm0.41}$ | $6.05_{\pm0.18}$ | $8.86_{\pm0.50}$ | $10.02_{\pm0.44}$ | $11.04_{\pm0.51}$ | $31.58_{\pm0.98}$ |
| | PropFair | $14.97_{\pm0.68}$ | $6.44_{\pm0.34}$ | $5.40_{\pm1.28}$ | $7.00_{\pm1.11}$ | $8.06_{\pm1.07}$ | $28.89_{\pm0.91}$ |
| | VRed | $20.42_{\pm0.36}$ | $6.08_{\pm0.05}$ | $9.43_{\pm1.01}$ | $11.21_{\pm0.74}$ | $12.81_{\pm0.85}$ | $33.59_{\pm1.11}$ |
| | Semi-VRed | $\mathbf{20.85}_{\pm0.39}$ | $6.26_{\pm0.18}$ | $9.12_{\pm1.47}$ | $\mathbf{11.86}_{\pm0.74}$ | $\mathbf{13.46}_{\pm0.63}$ | $\mathbf{34.57}_{\pm1.20}$ |
| CINIC-10 | FedAvg, Ditto | $86.26_{\pm0.03}$ | $15.20_{\pm0.07}$ | $50.48_{\pm0.29}$ | $56.87_{\pm0.36}$ | $62.78_{\pm0.16}$ | $100.0_{\pm0.00}$ |
| | $q$-FFL | $\mathbf{86.63}_{\pm0.06}$ | $\mathbf{14.88}_{\pm0.08}$ | $51.57_{\pm0.82}$ | $57.77_{\pm0.36}$ | $\mathbf{63.62}_{\pm0.18}$ | $\mathbf{100.0}_{\pm0.01}$ |
| | AFL | $86.45_{\pm0.12}$ | $15.10_{\pm0.11}$ | $51.57_{\pm0.45}$ | $57.58_{\pm0.29}$ | $63.04_{\pm0.28}$ | $100.0_{\pm0.00}$ |
| | GiFair | $86.28_{\pm0.11}$ | $15.20_{\pm0.13}$ | $49.66_{\pm1.21}$ | $56.97_{\pm0.29}$ | $62.74_{\pm0.36}$ | $100.0_{\pm0.00}$ |
| | TERM | $86.34_{\pm0.04}$ | $15.12_{\pm0.01}$ | $49.90_{\pm0.42}$ | $57.21_{\pm0.11}$ | $62.98_{\pm0.04}$ | $100.0_{\pm0.00}$ |
| | PropFair | $86.01_{\pm0.17}$ | $15.34_{\pm0.19}$ | $49.97_{\pm1.23}$ | $56.53_{\pm0.65}$ | $62.27_{\pm0.55}$ | $99.99_{\pm0.01}$ |
| | VRed | $85.79_{\pm0.35}$ | $15.02_{\pm0.06}$ | $51.57_{\pm0.50}$ | $57.66_{\pm0.30}$ | $62.75_{\pm0.36}$ | $99.98_{\pm0.01}$ |
| | Semi-VRed | $85.83_{\pm0.33}$ | $14.95_{\pm0.07}$ | $\mathbf{51.59}_{\pm0.98}$ | $\mathbf{58.00}_{\pm0.21}$ | $62.70_{\pm0.14}$ | $99.96_{\pm0.01}$ |
| StackOverflow | FedAvg, Ditto | $40.34_{\pm0.06}$ | $6.98_{\pm0.03}$ | $25.64_{\pm0.11}$ | $27.12_{\pm0.06}$ | $30.35_{\pm0.03}$ | $49.70_{\pm0.07}$ |
| | $q$-FFL | $37.79_{\pm0.80}$ | $7.38_{\pm0.09}$ | $22.54_{\pm1.03}$ | $24.12_{\pm1.00}$ | $27.14_{\pm0.92}$ | $47.06_{\pm0.66}$ |
| | AFL | - | - | - | - | - | - |
| | TERM | $40.34_{\pm0.05}$ | $6.96_{\pm0.06}$ | $25.56_{\pm0.21}$ | $27.12_{\pm0.20}$ | $30.41_{\pm0.12}$ | $49.76_{\pm0.10}$ |
| | GiFair | $40.34_{\pm0.04}$ | $6.97_{\pm0.03}$ | $25.71_{\pm0.13}$ | $27.10_{\pm0.11}$ | $30.34_{\pm0.08}$ | $49.71_{\pm0.09}$ |
| | PropFair | $41.76_{\pm0.01}$ | $6.80_{\pm0.05}$ | $27.30_{\pm0.21}$ | $28.75_{\pm0.19}$ | $32.14_{\pm0.10}$ | $50.76_{\pm0.08}$ |
| | VRed | $42.90_{\pm0.05}$ | $6.64_{\pm0.01}$ | $29.08_{\pm0.09}$ | $\mathbf{30.39}_{\pm0.05}$ | $33.55_{\pm0.05}$ | $51.66_{\pm0.03}$ |
| | Semi-VRed | $\mathbf{42.90}_{\pm0.03}$ | $\mathbf{6.60}_{\pm0.01}$ | $\mathbf{29.10}_{\pm0.06}$ | $30.34_{\pm0.09}$ | $\mathbf{35.55}_{\pm0.05}$ | $\mathbf{51.70}_{\pm0.04}$ |

## C.4 RELATION BETWEEN VRED AND ROBUST OPTIMIZATION

Empirical optimization is usually used as a data-driven approach for tuning models for decision making, where an expected loss is minimized based on some available train data. The trained model is then used for prediction tasks on some test data. However, if the empirical distribution of the train data is substantially different from that of test data, our confidence for doing prediction on the test data with the trained model diminishes. Robust empirical optimization has been used to address this problem (Bertsimas et al., 2018b;a; Ben-Tal et al., 2013). The work in (ya Gotoh et al., 2018) formulated a distributionally robust optimization (DRO) problem based on a minimax problem, where a model is trained on the given train data against the worst distribution shifts between the train and test data:

$$\min_{\theta} \max_{\mathbb{Q}} \left\{ \mathbb{E}_{(x,y)\sim\mathbb{Q}}[\ell(h(x,\theta),y)] + \frac{1}{\delta}\mathcal{H}_{\phi}(\mathbb{Q}|\hat{\mathbb{P}}_n) \right\}, \tag{42}$$

, where $\hat{\mathbb{P}}_n$ and $\mathbb{Q}$ are the train data empirical distribution and the test data distribution. The above problem optimizes against the "worst-case" test distribution $\mathbb{Q}$. The deviation of the distribution $\mathbb{Q}$ from $\hat{\mathbb{P}}_n$ is penalized in the regularization term $\frac{1}{\delta}\mathcal{H}_\phi(\mathbb{Q}|\hat{\mathbb{P}}_n)$, where $\mathcal{H}_\phi$ is a divergence measure between two distributions. The solution to this optimization problem is a model which is robust against distribution shifts between the train and test data. It was shown in (ya Gotoh et al., 2018) [see Propositions 3.1 and 3.2] that the above DRO problem is equivalent to a mean-variance problem, where the empirical average loss on train set is regularized with sample variance of the loss:

$$\min_\theta \max_\mathbb{Q} \left\{ \mathbb{E}_{(x,y)\sim\mathbb{Q}}[\ell(h(x,\theta),y)] + \frac{1}{\delta}\mathcal{H}_\phi(\mathbb{Q}|\hat{\mathbb{P}}_n) \right\} \equiv$$

$$\min_\theta \left\{ \mathbb{E}_{(x,y)\sim\hat{\mathbb{P}}_n}\left[\ell(h(x,\theta),y)\right] + \frac{\delta}{2\phi''(1)} \mathbb{E}_{(x,y)\sim\hat{\mathbb{P}}_n}\left[\ell(h(x,\theta),y) - \mathbb{E}_{(x,y)\sim\hat{\mathbb{P}}_n}[\ell(h(x,\theta),y)]\right]^2 \right\}. \tag{43}$$

This means that variance regularization is equivalent to DRO and can improve out-of-sample (test) performance. Maurer & Pontil (2009); Namkoong & Duchi (2017) propose regularizing the empirical risk minimization (ERM) by the empirical variance of losses across training samples to balance bias and variance and improve out-of-sample (test) performance and convergence rate. Similarly, Shivaswamy & Jebara (2010) propose boosting binary classifiers based on a variance penalty applied to exponential loss.

DRO is also an effective approach to deal with imbalanced and non-iid data. Unlike the above sample-wise variance regularization works, the work (Krueger et al., 2021) - assuming having access to data from multiple training domains - proposes penalizing variance of training risks across the domains as a method of distributionally robust optimization to provide out-of-distribution (domain) generalization. The first work propopsing DRO in `FL` setting is (Mohri et al., 2019), where they minimize the maximum combination of clients local losses to address fairness in FL:

$$\min_\theta \max_i f_i(\theta). \tag{44}$$

Also, the work in (Deng et al., 2020) proposea distributionally robust `FL` by minimizing the worst combination of clients local losses via periodic averaging and adaptive sampling:

$$\min_\theta \max_{\boldsymbol{\lambda}\in\Lambda} \sum_{i=1}^n \lambda_i f_i(\theta), \tag{45}$$

where $\boldsymbol{\lambda} \in \Lambda = \{\lambda \in \mathbb{R}_+^n : \sum_{i=1}^n \lambda_i = 1\}$. In contrast, our proposed `VRed` penalizes the variance of losses across clients for improving fairness (performance consistency) in `FL` settings with high data heterogeneity:

$$\min_\theta \sum_{i=1}^n \lambda_i f_i(\theta) + \beta \sum_{i=1}^n \lambda_i \left( f_i(\theta) - \sum_{j=1}^n \lambda_j f_j(\theta) \right)^2, \tag{46}$$

where $\lambda_i = \frac{n_i}{N}$, is the sample size of client $i$ and is fixed. The relation between robust optimization and variance regularization in non-`FL` settings (eq. (43)) encourages us to interpret `VRed` as an equivalent form of DRO. Hence, although the variance regularization used in `VRed` connects it non-trivially to the previous works `AFL` (Mohri et al., 2019) and `DRFA` (Deng et al., 2020) through DRO, it does not use a minmax objective function with potential convergence problems. As we have reported in our experiments, `AFL` fails to converge in settings with high data heterogeneity. Similarly, the authors of (Deng et al., 2020) have evaluated the `DRFA` algorithm only on Logistic Regression model, which is a convex problem. Furthermore, as reported in our results (and also reported in (Deng et al., 2020), fig. 3) `AFL`, thanks to its DRO formulation, can improve the fairness

(performance consistency) in the system. However, it clearly degrades the overall average performance. Similarly, `DRFA` (as reported in fig. 3 of (Deng et al., 2020)) can improve the system fairness keeping the same level of global accuracy as `FedAvg`. This is our motivation for proposing our `Semi-VRed` algorithm by solving the following problem instead of the previous DRO-based algorithms:

$$
\min_{\theta} \ \sum_{i=1}^{n} \lambda_i f_i(\theta) + \beta \sum_{i=1}^{n} \lambda_i \left( f_i(\theta) - \sum_{j=1}^{n} \lambda_j f_j(\theta) \right)_{+}^{2} ,
\tag{47}
$$

where $\lambda_i = \frac{n_i}{N}$, is the sample size of client $i$ and is fixed. We have shown by lemma 2 and lemma 3 and also example 1 that our `Semi-VRed` has a smarter and more efficient formulation for achieving fairness in `FL` systems, which results in improvement of fairness without degrading the system overall performance.

