# OpenReview forum: "Semi-Variance Reduction for Fair Federated Learning"
_ICLR.cc/2023/Conference — Submitted to ICLR 2023_

### Official Review · Reviewer_bZXJ · 2022-10-22

**Confidence:** 3
**Correctness:** 3
**Technical Novelty And Significance:** 3
**Empirical Novelty And Significance:** 3
**Recommendation:** 6

**Clarity, Quality, Novelty And Reproducibility:**

Regarding the clarity, I must point out two issues:

- There should be some fixes to Algorithm 1: The parameter $\theta$ appears multiple time in the algorithm, but it is not clear what it is; is it $\theta_t$, or is it an adapted version, or maybe even a parameter of the previous global epoch?

- The notion of fairness under consideration should be explicitly spelled out. From the experimental results, I guess that the ultimate aim of this work is to maximize both (1) the worst-client accuracy, and (2) the average-client accuracy. Is this true? If so, it should help readers a lot to clarify this point early in the text.

Other than that, I do think that this manuscript is of sufficient quality, novelty, and reproducibility.

**Strength And Weaknesses:**

__Strengths__

- Theorem 1 seems to be a working way of giving convergence guarantees for variance-penalized optimizations under the context of federated learning. The proof idea may be useful for the future works that aim to give convergence guarantees based on alternative risk measures on clients (other than averaging).

- I like how authors compared the empirical performance of the proposed algorithm with a variety of baselines on the experimental setups that, up to my knowledge, has not been popularly used in the literature. From the appendices, I see that authors have put considerable effort in tuning the hyperparameters of the baseline methods, which will be a great academic asset of the community if the code is released.

__Weaknesses__

- It is not clear to me why authors did not consider any model personalization in this context. In footnote 1, authors explain that authors do not use model personalization in this work for a fair comparison with baseline algorithms. Putting a constraint purely for a fair comparison is not a very practical thing to do---if one can have improved (average and worst-case) performance by removing a constraint, why should one even consider a setup with such constraint? Could you provide any practical justification on why it is (at least sometimes) useful to consider federated learning without any model personalization?

- The performance of the trained models seems to be very low overall, which may be due to an excessive number of clients. For instance, the CIFAR-10 performance with ResNet-18 models is quite low; lower than 50% accuracy. It may be useful to have comparisons under such regime, but it will also be useful to have experimental results in a more well-performing regime, where the computation/communication budget is higher and the number of clients is perhaps smaller.

- The discussion about Theorem 1 is rather weak. Is there any helpful knowledge that we could deduce from the upper bound of Theorem 1? Comparing with the discussion on lemma 2, I think that Theorem 1 is way more non-trivial than lemma 1,2 should be treated better.

**Summary Of The Paper:**

This paper provides two federated learning algorithms that aim to minimize the gap between the average performance of the trained model among clients and the performance on the worst-performing subgroups. The methods are designed to either minimize the variance or semi-variance of the clients' performance; called VRed and Semi-VRed, respectively. Authors establish convergence guarantees for VRed, and give several lemma to demystify the proposed algorithms. Through experiments, authors show that the performance of the proposed algorithms, especially that of Semi-VRed, is better than the prior state-of-the-arts.

**Summary Of The Review:**

This paper gives a simple, concrete, and well-performing federated learning algorithm with a sufficient degree of empirical validation. The paper could be much more impactful with some added experiments, but I do not see a very big reason to reject the paper.

---

> ### Author Response · Authors · 2022-11-16
> **Response to Reviewer bZXJ**
>
> Thank you for your critical comments. We provide the answers as follows.
>
> > **Q1:** It is not clear to me why authors did not consider any model personalization in this context.
>
> **A1:** The mechanism of model personalization algorithms for achieving fairness is further tuning of the global model based on the clients local data, which needs an extra computation budget on clients side. The other baselines mostly revolve around algorithmic innovation with no personalization and extra computation on clients side. Based on section 3.1 in its paper, Ditto has nothing more than FedAvg in its formulation except this extra computation (local personalization). So if we want to compare Ditto with the SemiVRed and the other baseline algorithms, we need to do this comparison in a fair way and that is why we have not considered the personalization for Ditto in our experiments. This simply reduces Ditto to FedAvg.
>
>
> > **Q2:** he performance of the trained models seems to be very low overall, which may be due to an excessive number of clients. For instance, the CIFAR-10 performance with ResNet-18 models is quite low; lower than 50% accuracy.
>
> **A2:** The reason behind the lower performance is the high data heterogeneity in our data distributions. As can bee seen in Table 2 in the appendix, we have set the Dirichlet parameter to 0.05 for CIFAR10 and CIFAR100 in order to have a high data heterogeneity across clients. We have stated in our paper that our proposed algorithms outperform the baseline algorithms with larger margins when the data is highly heterogeneous. We have also provided Example 1 in the appendix to provide some intuition why this is the case. Therefore, the large data heterogeneity that we consider is the reason why the trained model performance is relatively low.
>
> > **Q3:** The discussion about Theorem 1 is rather weak.
>
> **A3:** We have addressed your comment on our proof. We have written a new proof (in blue) for our convergence guaranties.
>
>
> > **Q4:** There should be some fixes to Algorithm 1: The parameter θ appears multiple time in the algorithm.
>
> **A4:** Thanks for reminding us about this typo. We had missed a subscript $t$, which we have written now in blue.
>
> > **Q5:** The notion of fairness under consideration should be explicitly spelled out.
>
> **A5:** The ultimate goal of this paper is to achieve fairness (consistency in clients text accuracy) without sacrificing the overall average performance (mean test accuracy). There have been multiple metrics proposed in the literature for measuring fairness (performance consistency) in FL settings, e.g. standard deviation of clients test accuracies (std), the worst client test accuracy, the worst 10\% and
> worst 20\% clients test accuracies. So in Table 6, we have reported the mean test accuracy (as the measure of the system overall performance) and also reported the mentioned fairness measures (as the measures of fairness in the system). We have shown that our proposed algorithms beat the baselines in terms of various fairness measures existing in the literature.

---

> > ### Comment · Reviewer_bZXJ · 2022-12-02
> > **Thank you for the reply.**
> >
> > Dear authors,
> >
> > Thank you for the reply. I have read the response, and other reviewers' comments.
> > I intend to keep my positive rating on the manuscript.
> >
> > Best,
> > Reviewer.

---

### Official Review · Reviewer_p3GJ · 2022-10-24

**Confidence:** 4
**Correctness:** 2
**Technical Novelty And Significance:** 2
**Empirical Novelty And Significance:** 3
**Recommendation:** 5

**Clarity, Quality, Novelty And Reproducibility:**

This paper is easy to follow. The motivation is very interesting and the proposed problem is novel. But it seems that the proposed method lacks of novelty, since the core idea is similar to reweight for each client. A deep discussion about the difference of the proposed method and the previous work is expected. The authors have provided code and I think the reproducibility can be guaranteed.

**Strength And Weaknesses:**

Strength:
1) This paper is well-motivated. Client-level performance fairness is an important topic in federated learning. Most of existing studies improve the fairness with the scarification of overall performance, while this paper focuses on both fairness and overall performance.
2) This paper is well-organized and the presentation is clear.

Weakness:
1) (Main Concern) I cannot understand why the proposed algorithm can protect the overall average performance from being sacrificed? According to the objective function Eq. (3) and Eq. (9), the proposed method improves fairness by minimizing the combination of overall risk and variance. Among them, the semi-VRed only penalizes the clients with local risk higher than the average risk. The semi-VRed does not encourage the clients with good performance to be closed to the average performance explicitly, however, it seems that there is also a performance sacrifice. There are always some clients that are up-weighted, which means that other clients are down-weighted relatively.
2) From the view of distributionally robust optimization, it seems that the proposed method is similar to reweight techniques (but with a smaller uncertainty set), which are widely used in the previous work, such as AFL [1] and DRFA [2]. The relations between this work and distributionally robust optimization-based methods should be discussed more.
3) The hyper-parameter local step number $K$ should be discussed, and it would be better to add more ablation studies about $K$, since there may be a trade-off between communication cost and performance (fairness). If $K$ is too small, more rounds of communication are needed to guarantee the convergence of federated model. If K is too large, the performance (fairness) may be not guaranteed, since the proposed federated algorithm is an approximation for the proposed objective function. Specifically, the difference between two rounds of local model parameters $f_i(\theta)-\bar{f}(\theta)$ is an unbiased estimation for the true weights (used to penalize the clients with worst performance) of clients, however, the variance of the estimation is very large (O(K^2)).

[1] Mohri M et. al. Agnostic federated learning. ICML 2019.
[2] Deng Yet. al. Distributionally robust federated averaging. NeurIPS 2020.


**Summary Of The Paper:**

This paper aims to guarantee the performance fairness among different clients in federated learning, while protecting the overall average performance from being sacrificed. The authors propose two algorithms based on variance reduction (in terms of all clients) and semi-variance reduction (in terms of the worst-off clients), respectively. Experimental results show that the proposed method can achieve SOTA performance.

**Summary Of The Review:**

This paper investigates a well-motivated problem, and the experimental results show the effectiveness of the proposed method.

However, I have some concerns about the technically soundness of the proposed methods. A deep discussion is expected.

---

> ### Author Response · Authors · 2022-11-16
> **Response to Reviewer p3GJ**
>
> Thank you for your feedback and questions. We are happy to answer your questions:
>
> > **Q1**: I cannot understand why the proposed algorithm can protect the overall average performance from being sacrificed?
>
> **A1**: We draw your attention to equation 12 for SemiVRed (compared to equation 8 for VRed), Remark 1 and also Table 1 in our paper. Thanks to its efficient formulation, SemiVRed assigns the weights to the best clients based on how well the worst ones perform. As the worst ones performance improve gradually, SemiVRed assigns larger weights to the best ones and lets them improve (equation 12), while this is not the case for VRed and the other baseline algorithms: the better a client performs, the more they suppress it. Although SemiVRed down-weights the good clients to some extent, it gradually reduces this down-weighting as the worst clients improve. In fact, in order to achieve performance consistency across clients, VRed and the other baseline algorithms 1. improve the worst clients performance and 2. deteriorate the best ones performance. However, SemiVRed 1. improves the worst clients performance and 2. along with that reduces the down-weighting of the best ones and lets them improve gradually. Therefore, as table 1 shows, this results in 1. **improvement of the worst clients** 2. **much less suppression of best clients** (compared to the baselines). With these two happening, both the fairness and overall average performance improve.
>
> > **Q2**: It seems that the proposed method is similar to reweight techniques (but with a smaller uncertainty set), which are widely used in the previous work, such as AFL and DRFA.
>
> **A2**: We have added section C.4 to the appendix explaining about the relation between VRed, which uses variance regularization, and the Distributionally Robust Optimization (DRO) algorithms AFL and DRFA. The relation between robust optimization and variance regularization in non-FL settings  encourages us to interpret VRed as an equivalent form of DRO. Although the variance regularization used in VRed relates it to DRO non-trivially, it has some superiorities over AFL and DRFA. It does not use a minimax objective function with potential convergence problems, as AFL and DRFA do. As we have reported in our experiments, AFL fails to converge in settings with high data heterogeneity. Similarly, the authors of DRFA have evaluated the DRFA algorithm only on Logistic Regression model, which is a convex problem. Furthermore, as reported in our results (and also reported in fig.3 of DRFA paper) AFL and DRFA can improve the fairness (performance consistency) in the system, thanks to their DRO formulation. However, they clearly degrade the overall average performance, or keep it on the same level as FedAvg.
>
>
> > **Q3**: The hyper-parameter local step number K should be discussed, and it would be better to add more ablation studies about K.
>
> **A3** The first point is that, as we are using SGD (with a default local batch size 64), the models are trained locally at each client to a good extent. So we expect increasing the parameter $K$ not to have much effect on the performance. If we were using larger values of batch size, $K$ might be affective by further local training. We ran a set of experiments on the most competing algorithms with SemiVRed on StackOverflow, i.e. FedAvg, PropFair, VRed (see table 6 in the appendix, where $K=1$) with $K=5$ to investigate its effect. The following shows the results:
> | algorithm | Mean  | Std  | Worst| Worst (10\%)  | Worst (20\%)  | Best (10\%) |
> |---|---|---|---|---|---|---|
> |FedAvg|  $41.75_{\pm 0.02}$ | $6.74_{\pm 0.03}$ | $27.26_{\pm 0.05}$ | $28.93_{\pm 0.02}$  | $32.07_{\pm 0.04}$  | $50.53_{\pm 0.04}$ |
> | PropFair|  $42.16_{\pm 0.03}$  | $6.74_{\pm 0.05}$  | $27.52_{\pm 0.04}$  | $29.32_{\pm 0.06}$  | $32.69_{\pm 0.04}$  | $50.45_{\pm 0.03}$ |
> | VRed| $42.11_{\pm 0.02}$ | $6.79_{\pm 0.04}$  | $27.36_{\pm 0.05}$  | $29.04_{\pm 0.03}$  | $32.37_{\pm 0.02}$ | $50.58_{\pm 0.08}$ |
> | SemiVRed|  **$\bf42.36_{\pm 0.04}$** | **$\bf6.70_{\pm 0.06}$** | **$\bf27.86_{\pm 0.03}$** | **$\bf29.50_{\pm 0.07}$** | **$\bf32.91_{\pm 0.05}$** | **$\bf50.83_{\pm 0.02}$** |
>
> As can be observed, SemiVRed still outperforms all the other baselines, improving both fairness and overall performance.

---

### Official Review · Reviewer_uqhW · 2022-10-25

**Confidence:** 3
**Correctness:** 3
**Technical Novelty And Significance:** 3
**Empirical Novelty And Significance:** 3
**Recommendation:** 6

**Clarity, Quality, Novelty And Reproducibility:**

The paper is well-written, and the proposed algorithm is novel based on my knowledge.

**Strength And Weaknesses:**

Strength:
The proposed algorithm has a very intuitive justification, i.e., improving the overall performance without sacrificing the utilities of well-performing clients by penalizing the discrepancy of
only the worst-off client’s loss functions from the average loss. The connection between the proposed algorithms and risk modeling methods in Finance is also interesting. The experimental results also validate the proposed algorithm.

Weaknesses:
The convergence result in Theorem1 is really hard to digest. It only provides an upper bound for the minimum expected gradient norm, and the second term will not converge to zero as T increases. It cannot show that the proposed algorithm will converge, and it is purely a bound for empirical risk during the optimization process instead of a bound of population risk or generalization error, which is not that informative in practice.


**Summary Of The Paper:**

This paper proposes two new fair federated learning (FL) algorithms, Variance Reduction (VRed) and Semi-Variance Reduction (Semi-VRed), which are inspired by two well-known risk modeling methods in Finance. VRed encourages equality between clients loss functions by penalizing their variance. In addition, Semi-VRed penalizes the discrepancy of only the worst-off clients loss functions from the average loss. Through extensive experiments on multiple vision and language datasets, it is shown that Semi-VRed achieves SoTA performance in scenarios with highly heterogeneous data distributions and improves both fairness and system overall average performance.

**Summary Of The Review:**

The proposed algorithm is easy to implement and effective based on the experiments provided in the paper. However, there is still room for improving the theoretical guarantee in section 5.

---

> ### Author Response · Authors · 2022-11-16
> **Response to Reviewer uqhW**
>
> Thank you for your feedback and we would like to answer your question as below.
>
> > **Q1**: The convergence result in Theorem1 cannot show that the proposed algorithm will converge
>
> **A1**: We have addressed your comment on our proof. We have written a new proof (in blue) for our convergence guaranties.

---

### Official Review · Reviewer_YTzx · 2022-10-27

**Confidence:** 3
**Clarity, Quality, Novelty And Reproducibility:** n/a
**Correctness:** 2
**Technical Novelty And Significance:** 2
**Empirical Novelty And Significance:** 2
**Recommendation:** 3

**Strength And Weaknesses:**

-- I am surprised that the most closely related paper is not even mentioned. It seems clear that the authors are not aware that similar risk measures, adopted from mathematical finance, have been proposed in the group fairness literature.  In particular, conditional value at risk (CVaR) has been used in the literature since the following seminal work.

https://proceedings.mlr.press/v97/williamson19a.html

Though it's not exactly the variance form, I believe that the CVaR formulation will be very similar when applied in the federated learning setting. The authors should have discussed the similarities/differences between the proposed approach and the CVaR formulation + FedAvg approach.

-- Also, it will be great if the authors can comment on the relationship between robust optimization and mean/variance optimization. According to [Gotoh, Kim, Lim], they are essentially equivalent. This means that the robust federated learning algorithm such as DITTO might be very similar to what the authors proposed.

https://www.sciencedirect.com/science/article/pii/S016763771730514X

-- Some baselines are missing: Addressing Algorithmic Disparity and Performance Inconsistency in Federated Learning, NeurIPS'21

**Summary Of The Paper:**

The authors propose a new fair FL algorithm that penalizes the variance (or semi-variance) of the utilities among the clients.

**Summary Of The Review:**

See my comments above.

---

> ### Author Response · Authors · 2022-11-16
> **Response to Reviewer YTzx**
>
> Thank you for your feedbacks and comments about our work. We are happy to answer your questions:
> >  **Q1**: Similar risk measures, adopted from mathematical finance, have been proposed in the group fairness literature. In particular, conditional value at risk (CVaR) has been used in the literature
>
> **A!**: The first point about the mentioned work is that the notion of fairness in their work (group fairness, as opposed to utility consistency in our work) and their motivation behind proposing the $\texttt{CVaR}_\alpha$ objective is totally different from those of our proposed SemiVRed algorithm: the authors clearly mention in their abstract that they propose a new notion of fairness and a new fairness-aware convex objective (based on $\texttt{CVaR}_\alpha$) to address the shortcomings of the previously proposed  fairness definitions in: 1. handling categorical/real-valued sensitive attributes 2. resulting in non-convex objective functions. In contrast, our motivation is to achieve fairness in FL settings (performance consistency) without sacrificing the overall performance. We are the first to adopt the Semi-Variance risk measure to FL settings to achieve this goal.
>
> The second point is that we cannot achieve the mentioned goal by adopting their proposed $\texttt{CVaR}_\alpha$ objective to our FL setting. We draw your deep attention about the subtle point that, although the $\texttt{CVaR}_\alpha$ measure models the one-sided behavior of a random variable, as the authors have shown in equations 22, 23 and 24 in the work, for $\alpha=0$, $\texttt{CVaR}_\alpha$ reduces to average of subgroups risks. However, for $\alpha>0$, $\texttt{CVaR}_\alpha$ reduces to the equation 24 in their paper, which is an **average of the largest subgroup risks** and completely ignores the smallest ones (similar to AFL). Hence, unlike our SemiVRed objective, for $\alpha>0$, their proposed $\texttt{CVaR}_\alpha$ objective function results in a trade-off between utility (accuracy) and fairness, as the authors have clearly shown in figure 1 in their paper. This is exactly what SemiVRed tries to avoid. Hence, our work is the first work addressing the fairness-utility trade-off in FL systems using the semi-variance risk measure.
>
> >  **Q2**: it will be great if the authors can comment on the relationship between robust optimization and mean/variance optimization
>
> **A2**: We have added section C.4 to the appendix explaining about the relation between VRed, which uses variance regularization, and Distributionally Robust Optimization (DRO) algorithms. Variance regularization has been used as an equivalent form of robust optimization for dealing with out-of-sample performance improvement and domain generalization. It is well-known that there is usually a trade-off between robustness and utility (accuracy). This is why AFL and VRed, which has roots in robust optimization, achieve robustness and fairness with the cost of a degradation in the system overall average performance. It is also clearly mentioned in section 3.1 of Ditto paper that it suffers from a trade-off between robustness/fairness, for which it is designed, and accuracy. Hence, due to this inefficiency of the aforementioned fair FL algorithms, we proposed SemiVRed to improve both the fairness and overall performance at the same time. Also, note that Ditto is designed for robustness against **data and model poisoning attacks**, and for that it uses model personalization with the cost of an extra local optimization problem on each client side, which our proposed algorithms do not require, and this makes Ditto different from our SemiVRed.
>
> >  **Q3**: Some baselines are missing: Addressing Algorithmic Disparity and Performance Inconsistency in Federated Learning, NeurIPS'21
>
> **A3**: The authors of the mentioned work try to achieve two notions of fairness at the same time: 1. performance consistency (the fairness notion which we aim to achieve in our work) 2. algorithmic (group) fairness. As shown in equation 8 in the work, they try to achieve these by solving a constrained minimax optimization problem, in which the objective function is the largest loss across clients and the constraints ensure local group fairness at each client. Now, we draw your attention to observe that the problem in equation 8 is a constrained version of AFL algorithm. Due to these constraints, and as clearly mentioned below equation 2 in the paper, the performance inconsistency will be magnified when the model is adjusted to satisfy group fairness on local clients. Hence, with respect to performance consistency notion of fairness, which we care about in our work, their algorithm cannot beat AFL. Therefore, it suffices to use AFL as a baseline in our experiments.

---

### Author Response · Authors · 2022-11-21
**Paper revision submitted**

Dear reviewers,

We, the authors of paper 2871, would like to thank you all for your constructive and valuable feedbacks about our work. We have addressed your comments and made relevant changes in our draft in blue. We hope our responses address your concerns. Should you have any further questions/concerns, please let us know about them. Thank you.

---

### Decision · Program_Chairs · 2023-01-20

**Decision:**

Reject

**Justification For Why Not Higher Score:**

See the aforementioned main concerns (A), (B), and (C).

**Justification For Why Not Lower Score:**

N/A.

**Metareview: Summary, Strengths And Weaknesses:**

The main contribution of this work lies in exploiting the Variance Reduction and Semi-Variance Reduction of clients' utilities for achieving equality in federated learning.

After reviewing the authors' rebuttal and an active discussion, the reviewers agree that the problem is well-motivated and interesting.

The main concerns among the reviewers are that (A) there is a lack of discussion of the novelty of this work relative to existing bodies of works (e.g., DRO, risk measures, reweight techniques); and (B) the initial convergence results do not show algorithm convergence. The authors have responded to both these concerns by (A) discussing the above-mentioned bodies of related work, and (B) providing a new proof of convergence guarantee. However, for (A), some reviewers remain concerned about the similarity of this work to CVaR + FL and reweight techniques. For (B), it is noted that a strongly convex assumption is required for the new proof. (C) Finally, it is not clear if it is practical to achieve fairness in the case of low performance of trained models on CIFAR-10/100, albeit due to high data heterogeneity.

The authors are strongly encouraged to revise their work based on the reviewers' feedback and suggestions.